# Understanding Complexity in VideoQA via Visual Program Generation

**Cristóbal Eyzaguirre** [1]   **Igor Vasiljevic** [2]   **Achal Dave** [2]   **Jiajun Wu** [1]   **Rares Andrei Ambrus** [2]   **Thomas Kollar** [2]
**Juan Carlos Niebles** [1]   **Pavel Tokmakov** [2]

ceyzaguirre4.github.io/codeplexity

## Abstract

We propose a data-driven approach to analyzing query complexity in Video Question Answering (VideoQA). Previous efforts in benchmark design have relied on human expertise to design challenging questions, yet we experimentally show that humans struggle to predict which questions are difficult for machine learning models. Our automatic approach leverages recent advances in code generation for visual question answering, using the complexity of generated code as a proxy for question difficulty. We demonstrate that this measure correlates significantly better with model performance than human estimates. To operationalize this insight, we propose an algorithm for estimating question complexity from code. It identifies fine-grained primitives that correlate with the hardest questions for any given set of models, making it easy to scale to new approaches in the future. Finally, to further illustrate the utility of our method, we extend it to automatically generate complex questions, constructing a new benchmark that is 1.9 times harder than the popular NExT-QA.

## 1. Introduction

Humans can effortlessly reason about activities, whether that reasoning requires understanding space and time, cause and effect, or fine-grained details and high-level context (Decety & Grèzes, 1999; Decety et al., 1997; Wurm & Caramazza, 2022; Aflalo et al., 2020). This versatility allows us to function effectively in dynamic environments, yet it simultaneously complicates our ability to assess what is hard for machines. Consider the two video-question pairs

shown in Figure 1 (top). In our study, human subjects overwhelmingly perceive the question on the right as the more complex to answer, but evaluating a variety of state-of-the-art VideoQA models (Yu et al., 2023a; Wang et al., 2022a; Surís et al., 2023; Fu et al., 2021) shows that the question on the left presents a significantly greater challenge for them.

Experts might assume they would perform better at this task, but the history of VideoQA benchmarks suggests otherwise. Despite authors' best efforts, studies show most datasets are dominated by questions solvable by naive, single-frame baselines (Buch et al., 2022; Huang et al., 2018; Liu et al., 2021). Although many attempts have been made to address this limitation, they predominantly adopt a top-down approach. These works start from an expert hypothesis of what is hard and validate this assumption by evaluating models on samples that specifically target the identified skill (Xiao et al., 2021; Mangalam et al., 2023). While this has led to some progress, such static, heuristic-based definitions of complexity are inherently myopic.

In this work, we propose a bottom-up approach instead that discovers human-interpretable insights about the sources of complexity for VideoQA models from the data. To this end, we capitalize on a recent large language model (LLM)-based code generation paradigm (Surís et al., 2023; Gupta & Kembhavi, 2023; Subramanian et al., 2023), which produces modular executable programs to answer natural language queries. While this approach has shown promise for zero-shot VideoQA (Surís et al., 2023; Ge et al., 2024), we are not interested in its task performance per se. Instead, we use its rich, structured intermediate representations—programs, as shown in Figure 1 (bottom)—to capture the elusive complexity of the original questions.

Specifically, we begin by collecting visual programs generated by recent methods (Surís et al., 2023; Ge et al., 2024) on the validation set of the challenging NExT-QA benchmark (Xiao et al., 2021), together with predictions of a large collection of diverse algorithms. We then calculate several standard structural complexity metrics (McCabe, 1976) for these programs and collect human judgments for a subset of the dataset. Intriguingly, our analysis demonstrates that, despite the programs being imperfect,

---

[1]Stanford Computer Science [2]Toyota Research Institute. Correspondence to: Cristóbal Eyzaguirre <ceyzaguirre@cs.stanford.edu>.

*Proceedings of the 42nd International Conference on Machine Learning*, Vancouver, Canada. PMLR 267, 2025. Copyright 2025 by the author(s).

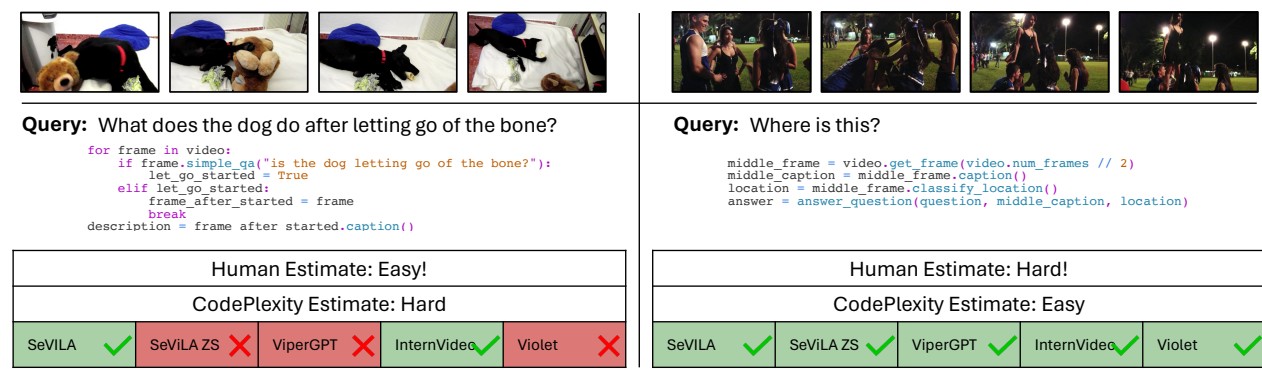

Figure 1. Humans struggle to judge which questions present higher challenges for machine learning models. In our study, the question on the left is universally perceived as being easier than the one on the right, which is inversely correlated with the models' performance. We show that the complexity of the corresponding visual program can serve as a much more reliable predictor.

standard code complexity metrics correlate better with machine learning models' performance than human estimates (see Figure 3). Moreover, improvements in code generation result in stronger corelation (Figure 19), and these observations generalize between benchmarks (Figure 20).

We then propose CodePlexity – a novel algorithm for estimating question complexity from code that takes into account the content of the program, in addition to its structure (Section 3.2). In particular, it learns to correlate individual subroutines with models' performances, effectively mining for human-interpretable patterns that summarize the error modes of any given set of models. Crucially, like any data-driven approach, our CodePlexity metric is easy to extend to new models and datasets, ensuring sustained progress. This is in stark contrast to the static, heuristic-based definitions of complexity that are dominant in the field.

Finally, to further demonstrate the utility of our analysis tool, we design an algorithm for automatically generating challenging questions for any given collection of videos in Section 3.4. In particular, our approach takes as input a compact description of a video and uses an LLM (OpenAI, 2023a) to generate question candidates first. We then generate visual programs for each question and use our code-based metric to select the hardest subset. We evaluate several zero-shot VideoQA methods on the resulting benchmark and observe a $1.9\times$ gap in performance compared to existing datasets like NExT-QA (Xiao et al., 2021).

To summarize, our contributions are as follows: (1) We demonstrate that generated code complexity is a robust, data-driven metric of question complexity in VideoQA, generalizing across code generation methods and datasets; (2) We present CodePlexity, an adaptable tool for evaluating model-specific challenges in VideoQA, identifying common failure modes and providing actionable insights for future research; (3) We use CodePlexity to automatically construct CodePlex-QA — a novel benchmark that is 1.9 times harder than the popular NExT-QA.

## 2. Related Work

**Large single-stage models.** Current methods for video-language understanding explores various modeling strategies.HGA (Jiang & Han, 2020) uses Graph Convolutional Networks (GCNs) to align video and language. Merlot (Zellers et al., 2021) leverages ASR-captioned videos for self-supervised training. VIOLET (Fu et al., 2021) employs dVAE (Van Den Oord et al., 2017) for masked video-text pretraining, evaluated on VideoQA and text-to-video retrieval. Masked space-time autoencoders, such as Times-Former (Bertasius et al., 2021) and VideoMAE (Feichtenhofer et al., 2022), focus on action recognition. mPLUG-2 (Xu et al., 2023) unifies image- and video-language tasks with task-specific modules. More recently, large language model (LLM)-based approaches, such as Tarsier (Wang et al., 2024), LLaVA-NeXT (Liu et al., 2024), and VideoChat2 (Maaz et al., 2024), explore unified video-language modeling by extending the language model with visual adapters. Similarly, SeViLA (Yu et al., 2023a) follows this LLM-based paradigm but adds an additional stage for frame selection, extending the select-then-answer approach popularized by ATP (Buch et al., 2022).

**Code generation models.** Recent works address frame selection and question answering via *code generation*, leveraging text-to-code models like Codex (Chen et al., 2021). VisProg (Gupta & Kembhavi, 2023) decomposes natural language queries into compositional programs using zero-shot pretrained models. ViperGPT (Surís et al., 2023) generates and executes Python code via a vision-API prompt, achieving state-of-the-art results without additional training. CodeVQA (Subramanian et al., 2023), a concurrent work, specializes in single-frame QA with a smaller API. RVP (Ge et al., 2024) introduces a recursive strategy to break down complex queries into subproblems, enhancing flexibility. Earlier models (Andreas et al., 2016; Johnson et al., 2017b; Kim et al., 2018; Hu et al., 2017; Yi et al., 2018) integrated code generation through supervised learn-

ing or reinforcement learning.

**Complexity estimation.** Prior work has proposed ways to estimate complexity for other settings and modalities. In NLP, text complexity is often approximated by length (Platanios et al., 2019; Spitkovsky et al., 2010; Tay et al., 2019), with variations including conjunction count (Kocmi & Bojar, 2017), phrase count (Tsvetkov et al., 2016a), and dependency tree depth (Tsvetkov et al., 2016b). In computer vision, image complexity has been linked to object count (Wei et al., 2016) or human annotations (Tudor Ionescu et al., 2016; Soviany et al., 2020). Finally, (Graves, 2016) suggested that, in reasoning tasks, complexity can be estimated by the number of steps required by a recurrent model. Later (Eyzaguirre & Soto, 2020) used a similar approach it to quantify the complexity of VQA questions.

**Complexity from code.** Measuring complexity through code has a rich history; Kolmogorov defines complexity based on the succinctness of the program that can represent said object (Kolmogorov, 1963; Solomonoff, 2009). However, its incomputability limits its practical application (Zvonkin & Levin, 1970). Software engineering rely on tangible metrics like cyclomatic complexity that measure the number of independent paths in a program (McCabe, 1976), a computable yet less philosophically rich approach.

Synthetic datasets play a key role in Video Question Answering by using symbolic programs to separate perception from reasoning (Johnson et al., 2017a; Grunde-McLaughlin et al., 2021; Yu et al., 2023b; Wu et al., 2021). Grouping these programs into skill-based families enables correlation of model performance with reasoning patterns. In contrast, we leverage code generation to estimate question complexity without costly annotations, extending to diverse question types. Our approach provides a direct, computable complexity measure, bridging theoretical and practical aspects of machine learning.

# 3. Methodology

## 3.1. Preliminaries

We study the problem of estimating the complexity of questions in VideoQA. We are given a dataset consisting of collections of videos, questions, and answers $\mathcal{D} = \{\mathbf{V}, \mathbf{Q}, \mathbf{A}\}$, along with a set of $K$ models already trained on the task $\mathcal{M} = \{m_0, ..., m_K\}$. Our goal then is to design a function $\mathcal{C}$ that allows us to categorize questions $q_i \in \mathbf{Q}$ into groups based on their complexity with respect to $\mathcal{M}$. Crucially, we are interested in a general metric consistent across all models $m_j \in \mathcal{M}$. Concretely, for any two questions $q_1, q_2 \in \mathbf{Q}$, together with corresponding videos $v_1, v_2 \in \mathbf{V}$, if $\mathcal{C}(q_1) > \mathcal{C}(q_2)$, we expect model performance $P(m, q, v)$ to vary accordingly: $P(m_j, q_1, v_1) < P(m_j, q_2, v_2) \ \forall m_j \in \mathcal{M}$, indicating that models perform worse on more complex questions.

However, directly estimating complexity $\mathcal{C}$ from natural language question $q$ is a challenging problem even for humans, as we demonstrate in a Section 4.2. Instead, our key idea, inspired by the notion of Kolmogorov Complexity ($\mathcal{KC}$) (Kolmogorov, 1963), is to utilize the rich and highly-structured intermediate representations - programs, to capture the elusive complexity of the original natural language queries. Concretely, we capitalize on the recent code generation-based methods (Surís et al., 2023; Gupta & Kembhavi, 2023; Subramanian et al., 2023) that operate in a 2-stage fashion: first, given a question $q$ a program generator $\pi$ from a Large Language Model (LLM) is used to translate it into an executable program $z = \pi(q)$. An off-the-shelf execution engine like Python can then be used to produce an answer $\hat{a} = \phi(v, z)$. Running such an approach on a dataset $\mathcal{D}$ results in a set of programs $\mathcal{P}(\mathcal{D}) = \{z_1, z_2, ..., z_N\}$.

Next, in Section 3.2 we propose several techniques for code analysis of increasing intricacy and show how they can be used to build a function for question complexity estimation via code $\mathcal{C}(q) \propto \mathcal{C}(z)$. Then, in Section 3.3, we demonstrate how analysis of the generated code can help gain insights into the failure modes of VideoQA models. Finally, in Section 3.4 we discuss how such algorithms can be used to automatically construct challenging benchmarks.

## 3.2. CodePlexity: Estimating Question Complexity from Code

As a first step we review existing software engineering metrics that map code into complexity scores $\mathcal{C}(z) \to \mathbb{R}$. In particular, we focus on Lines of Code (LoC) and Cyclomatic Complexity (McCabe, 1976). The former simply correlates the number of lines in a program with its complexity $\mathcal{C}(z) \propto |z|$, whereas the latter quantifies the number of linearly-independent paths through the source code and is denoted as $\mathcal{C}(z) = CC(z)$. To minimize the impact of spurious factors, we pre-process the code by removing all the comments and empty lines first, and make sure to use the same set of basic primitives in all experiments. Both metrics are indicative of the code's structural complexity, with higher values suggesting more intricate control flow. However, they do not take the contents of the code into consideration, which, as we shown in Section 4.2, limits their predictive power.

To address this drawback, we propose a new method, Code-Plexity, illustrated in Figure 2. CodePlexity involves analyzing the components of the generated code that affect question complexity, considering both its structure and semantic content. More specifically, we develop a compiler to parse each question's code into its basic syntactic elements, creating a Abstract Syntax Tree or AST (Hoe et al.,

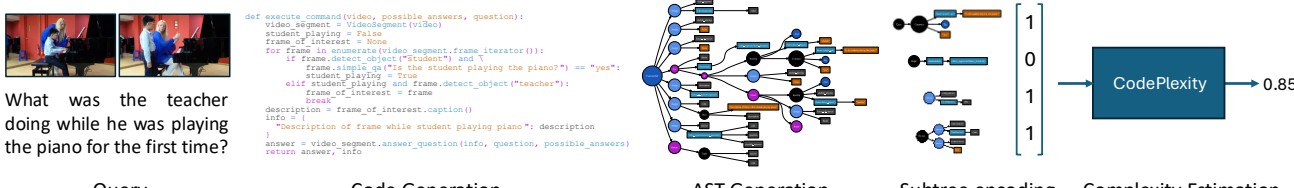

| Query | Code Generation | AST Generation | Subtree encoding | Complexity Estimation |

*Figure 2.* **Estimating question complexity via code.** Our approach to estimating question complexity involves converting the question into code, decomposing the pseudo-code into abstract syntax subtrees ($S_i$), before correlating subtree presence with model performance.

1986) $T = compile(z)$ with nodes $N$ and edges $E$. In this model, nodes represent variables, functions, and control structures, while the edges capture the logical and hierarchical relationships between them. The AST framework abstracts the code away from its syntax, allowing us to focus on the underlying logic and structure. By generating ASTs for the entire dataset, we obtain a comprehensive set $\mathcal{T}(\mathcal{D}) = \{compile(z) \mid z \in \mathcal{P}(\mathcal{D})\}$, laying the groundwork for a deeper analysis of code complexity factors.

Next, we mine $\mathcal{T}$ for common subroutines (recurring logical patterns or functions) that occur in the code. In ASTs subroutines manifest as subtrees, which we denote as $S = (N', E')$, where $N' \subseteq N$, $E' \subseteq E$ and $\forall (u, v) \in E'$, $u \in N' \land v \in N'$. Importantly, not all subtrees constitute valid Python code, since they might fail to comply with Python's syntax rules. To systematically identify valid subtrees, we define a function $\mathcal{G}(T)$ that yields an unordered set of all valid subtrees of $T$, denoted as $\mathcal{G}(T) = \{S_1, S_2, \ldots, S_n\}$. Considering the entire dataset, the collection of all valid subtrees across the dataset can be represented as $\mathcal{S}(\mathcal{D}) = \bigcup_{T \in \mathcal{T}(\mathcal{D})} \mathcal{G}(T)$.

Then, to avoid duplicates, we merge subtrees that always co-occur when one is a descendant of the other. Specifically, $\mathcal{S}_{merged}(\mathcal{D})$ is defined as the subset of $\mathcal{S}(\mathcal{D})$ that excludes $S_2$ if there exists a subtree $S_1$ such that $S_1$ and $S_2$ always co-occur and $S_2$ is contained in $S_1$ (see Section 7.1 for formal definition). The presence of a specific subroutine within a program's AST can be verified via a subgraph isomorphism check:

$$ISO(T, S) \equiv S \in \mathcal{G}(T). \tag{1}$$

To aggregate the identified subtrees into a quantitative metric of complexity, we assign each subtree in $\mathcal{S}_{merged}(\mathcal{D})$ an index and encode each question $q_i$ in the dataset using one-hot encoding $\mathbf{x}_i \in \mathbb{R}^{|\mathcal{S}_{merged}(\mathcal{D})|}$, where a 1 in index $k$ of $x_i$ signifies the presence of subtree $S_k$ in question's AST $T_i$.

$$x_{ik} = \begin{cases} 1 & \text{if } ISO(T_i, S_k) \\ 0 & \text{otherwise} \end{cases} \tag{2}$$

This representation transforms the complex structure of code into a fixed-size vector, enabling straightforward application of machine learning models. We then employ a

logistic regression model trained on these one-hot encodings to predict the success of models $m_j \in \mathcal{M}$. Note that the training set effectively treats each $(\mathbf{x}_i, y_i^{(j)})$ pair as a distinct instance, where $y_i^{(j)}$ is the binary outcome for question $i$ with respect to model $m_j$ (1 for success, 0 for failure). This approach is justified by our objective to identify subtrees that universally challenge the models, implying a structural complexity in the code that transcends specific models. We then obtain the final complexity function via:

$$\text{CodePlexity}(z) = -\hat{y}_i = -\sigma(\mathbf{w}\mathbf{x}_i + b). \tag{3}$$

Next, we discuss how our subtree analysis approach allows to obtain deeper insights into the sources of complexity for existing VideoQA models.

### 3.3. Subtree Analysis

Unlike black-box metrics, in addition to a numerical score, our approach also outputs an interpretable set of subtrees that correlate with challenging questions. We now demonstrate how to identify subroutines that have a high impact on model performance. More specifically, we are interested in subtrees that are linked to a decrease in model $m_j$'s performance with a high degree of statistical significance (set at 0.99). To test this, we establish a null hypothesis ($H0$) stating that the proportion of successes is the same with and without the subtree present:

$$H0 : P(m_j, q_1 | S \in \mathcal{S}(\mathcal{D})) = P(m_j, q_1 | S \notin \mathcal{S}(\mathcal{D})). \tag{4}$$

Conversely, our alternative hypothesis posits that the proportion of successes without the subtree is greater, implying that its presence hurts performance:

$$HA : P(m_j, q_1 | S \in \mathcal{S}(\mathcal{D})) < P(m_j, q_1 | S \notin \mathcal{S}(\mathcal{D})). \tag{5}$$

We conduct a one-sided test to evaluate these hypotheses and define a subset of subtrees, denoted as $\mathcal{S}^*_{m_j}(\mathcal{D})$, for which their presence is statistically correlated with a decrease in the performance of the model $m_j$:

$$\mathcal{S}^*_{m_j}(\mathcal{D}) = \{S \in \mathcal{S}(\mathcal{D}) \mid p(S, m_j) < 0.01\}, \tag{6}$$

where $p(S, m_j)$ denotes the corresponding p-value. Finally, to identify the subroutines that are associated with

performance decrease for multiple models, we consider the intersection of the sets:

$$\mathcal{S}^* = \bigcap_{m_j \in \mathcal{M}} \mathcal{S}^*_{m_j}(\mathcal{D}). \tag{7}$$

Identifying the specific subtrees that cause a decrease in models' performance allows us to obtain deeper insights into where and how they may falter. In Section 4.3, we perform this analysis for several state-of-the-art approaches and suggest areas for improvement in model design.

### 3.4. Learning to Ask Hard Questions

We now build on our code-based question complexity metric described in Section 3.2, and propose a method for automatically generating challenging question-answer pairs for any given set of videos. Concretely, our approach takes as input a set of videos $\mathbf{V}$ paired with natural language summaries $\mathbf{C}$. We then follow prior work by (Mangalam et al., 2023) and prompt a large language model (LLM) to generate question and answer candidates based on each summary individually $\tilde{q}, \tilde{a} = LLM(c, prompt)$. The exact prompts are listed in Section 7.4 of the appendix.

Importantly, our approach is agnostic to the nature of $\mathbf{C}$, which can either be annotated manually, or generated automatically. In this work, we take the latter approach and capitalize on existing datasets with scene graph annotations (Luo et al., 2021; 2022; Zhou et al., 2019; Ji et al., 2020) paired with an image captioning model to generate natural language summaries of the video such that a language model can understand them (Menon & Vondrick, 2022; Wang et al., 2022b; Zeng et al., 2023). We detail this algorithm in Section 8 of the appendix.

Following our approach from Section 3.2, we then convert each generated question $\tilde{q}$ into code, and use our trained CodePlexity model (Equation 3) to estimate its complexity. A set of candidate questions $\tilde{\mathbf{Q}}^*$ can be selected by setting a threshold $\delta$ for minimum complexity:

$$\tilde{\mathbf{Q}}^* = \{\tilde{q} \in \tilde{\mathbf{Q}} | \mathcal{C}(\tilde{q}) \geq \delta\}. \tag{8}$$

Finally, we manually filter the candidate dataset $\tilde{\mathcal{D}}^* = \{\mathbf{V}, \tilde{\mathbf{Q}}^*, \tilde{\mathbf{A}}^*\}$ to remove the question/answer pairs that cannot be accurately answered from the corresponding videos due to inaccuracies in the generated summaries or LLM hallucination. We emphasize that this final manual filtering is only needed to ensure the perfect quality of the final dataset $\mathcal{D}^*$. In practice, we only had to remove 12% of the questions, demonstrating that the fully automatic pipeline is capable of producing useful datasets by itself.

## 4. Evaluating Complexity Estimation

In this section, we compare different approaches to estimating question complexity in VideoQA. To this end, we first define a thorough evaluation protocol and detail our experimental setup in Section 4.1. We then evaluate how the code-based metrics proposed in this work compare to human subjects and several simple baselines in predicting the performance of a wide variety of contemporary approaches on the popular NextQA benchmark in Section 4.2. We conclude by performing a detailed analysis of the subroutines that show the strongest correlations with challenging questions in Section 4.3.

### 4.1. Experimental Setup

**Evaluation protocol.** Our goal is to compare the predictive power of several approaches for estimating question complexity in VideoQA with respect to a variety of machine learning models $\mathcal{M}$ on a dataset $\mathcal{D}$. Importantly, some of the metrics we study require training data in the form of questions paired with outcomes of a model $m_j \in \mathcal{M}$ on them $(q_i, y_i^{(j)})$. Thus we split the whole pool of models $\mathcal{M}$ into the training $\mathcal{M}_{tr}$ and held-out validation $\mathcal{M}_{val}$ sets and report results on the latter.

To quantitatively compare the effectiveness of different approaches, some of which map a question to a numerical value corresponding to its complexity, whereas others directly return an ordering of the questions, we propose a unifying metric, Performance Extremity Gap (PEG). In particular, we first use numerical complexity estimates to sort questions accordingly. We then measure the disparity in model $m_j$'s performance $P$ between the easiest and the hardest $\alpha\%$ of the questions via:

$$\begin{aligned} \text{PEG}(m_j, \alpha) = &\frac{1}{N_\alpha} \sum_{q \in Q_{\text{hardest}, \alpha}} P(m_j, q, v_q) \\ &- \frac{1}{N_\alpha} \sum_{q \in Q_{\text{easiest}, \alpha}} P(m_j, q, v_q) \end{aligned} \tag{9}$$

Finally, inspired by the mAP metric (Everingham et al., 2010), we average the PEG values over $\alpha \in (0, 0.5]$ to obtain the final mPEG score.

**CodeGen Models.** We leverage ViperGPT (Surís et al., 2023) as our main approach for generating visual programs from questions. To investigate the influence of the CodeGen model, we also re-ran our experiments with the recent RVP (Ge et al., 2024) approach (see Section 10.4).

**VideoQA Models.** We ground our complexity metric in the performance of seven representative VideoQA methods, chosen for their coverage of existing architectural philosophies, pre-training strategies, and state-of-the-art performance. In particular, we use VIOLET (Fu et al., 2021) and InternVideo (Wang et al., 2022a), which are pre-trained with contrastive visual-language objectives and fine-tuned for VideoQA. We also evaluate SeViLA (Yu et al., 2023a), which is based on the BLIP-2 (Li et al., 2023) large-scale

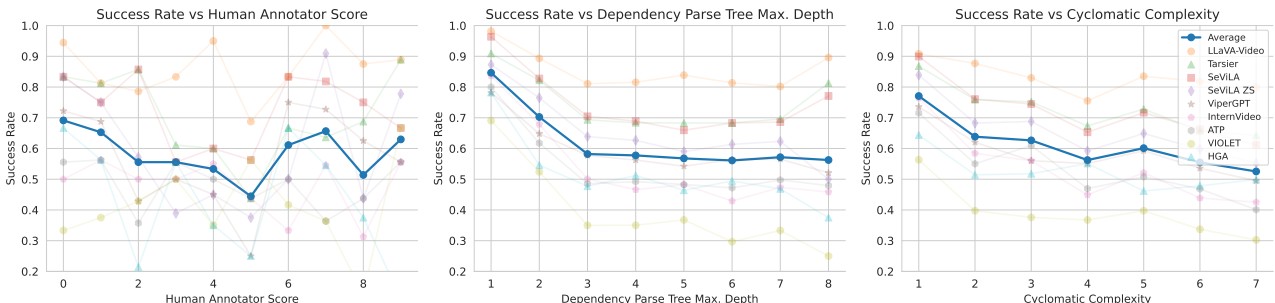

*Figure 3.* Correlation of various approaches for estimating question complexity with VideoQA models' success rate on these questions. We observe that humans struggle to accurately predict what is hard for machine learning models and that code can serve as a more reliable source of prediction than natural language questions.

visual-language model; we assess both its zero-shot variant (SeViLA-ZS) and a fine-tuned version (SeViLA). Additionally, we evaluate HGA (Jiang & Han, 2020), a GNN-based model for video reasoning, representing earlier approaches prior to the prevalence of video large language models (videoLLMs). We also evaluate the simple but effective ATP baseline (Buch et al., 2022), a model intentionally restricted to processing only a single frame. Furthermore, we include the results from executing the programs generated by ViperGPT (Surís et al., 2023). Finally, we evaluate the recent state-of-the-art model, Tarsier (Wang et al., 2024), and the current state-of-the-art model LLaVa-Video (Zhang et al., 2024). The models are split into training and validation sets as follows: $\mathcal{M}_{tr} = \{\text{VIOLET}, \text{SeViLA}, \text{ViperGPT}, \text{ATP}\}$, $\mathcal{M}_{val} = \{\text{HGA}, \text{SeViLA-ZS}, \text{InternVideo}, \text{Tarsier}, \text{LLaVa-Video}\}$.

**Dataset.** It is crucial that the dataset used to perform our analysis features as many diverse challenges as possible. We focus on the NExT-QA (Xiao et al., 2021) benchmark for its size, variety of human-annotated questions, and its focus on spatio-temporal reasoning in videos over mere visual-fact retrieval (Zhong et al., 2022). In addition, its popularity provides a large pool of models with pre-trained, public checkpoints for our study. We perform the evaluation on the validation set, further splitting the questions into 80% used to train the metrics and the other 20% held out for computing mPEG. To assess generalizability, we also evaluated our method on the recent MVBench dataset (Li et al., 2024), including additional models such as LLaVA-NeXT (Liu et al., 2024) and VideoChat2 (Li et al., 2024), in Section 10.5 of the appendix.

**Baselines.** In addition to the code-based metrics introduced in Section 3.2, we evaluate several baselines that attempt to directly estimate question complexity from the natural language query itself. In particular, as a learning-free baseline, we follow (Tsvetkov et al., 2016b) and correlate the complexity of a question with the maximum depth of its parsed dependency tree. To more fairly compare to our learnable, code-based metric we fine-tune BERT (Kenton

& Toutanova, 2019) to predict the probability of model success given the question using exactly the same training data. We also prompt GPT-4 (OpenAI, 2023b) to estimate the complexity of a question on a Likert scale (Likert, 1932). Details and prompts are provided in Section 7.3.

Finally, we conduct a human study on a subset of 150 questions. To this end, we recruited 30 human subjects via the Prolific platform (Palan & Schitter, 2018). The subjects were asked to sort three questions at a time according to their perceived relative complexity. The final sequence order of the entire subset was calculated via pairwise ELO scores (Elo, 1967). More details and an example of the annotation interface are provided in Section 7.2.

### 4.2. Results

We begin by visualizing the correlation of human estimates of question complexity with the performance of all 9 models used in our study on the manually annotated questions from NExT-QA in Figure 3 (left). We observe that, while a downward trend clearly exists, with the questions labeled as the hardest by humans resulting in lower success rate for models compared to the easiest ones, the correlation is very weak. Notably, the questions that are ranked as being average in complexity are in fact the hardest for the models.

We then evaluate two baselines on the same set of questions, one based on the natural language queries (dependency tree depth shown in Figure 3, center) and one based on the generated code (cyclomatic complexity, Figure 3, right). Both show a much stronger correlation with the models' performance, with cyclomatic complexity being the most consistent. These results demonstrate that human intuition about sources of complexity in VideoQA does not reflect the main challenges for machine learning models, and that generated code can be a more reliable source for estimating complexity than natural language.

Next, we report a more systematic comparison of different text- and code-based metrics using mPEG on the validation set of NExT-QA in Table 1. Comparing the three language-

|  | Train Models | | | | Val. Models | | | | |
|---|---|---|---|---|---|---|---|---|---|
|  | SeViLA | ViperGPT | ATP | VIOLET | HGA | SeViLA ZS | InternVideo | Tarsier | LLaVa-Video |
| Dependency Tree Depth | 12.9 | 7.9 | 11.1 | 15.9 | 7.4 | 13.5 | 17.7 | 10.1 | 6.9 |
| GPT-4 (OpenAI, 2023b) | 9.6 | 8.9 | 11.6 | 5.8 | 7.8 | 14.6 | 13.9 | 10.8 | 5.2 |
| BERT (Kenton & Toutanova, 2019) | 12.5 | 6.0 | 18.3 | 17.3 | 7.7 | 14.3 | 21.1 | 10.8 | 11.4 |
| Lines of Code | 16.4 | 15.3 | 14.2 | 12.0 | 9.9 | 16.2 | 17.5 | 14.4 | 9.38 |
| Cyclomatic Complexity | 18.2 | 14.2 | 18.7 | 15.9 | 8.9 | 17.2 | 24.2 | 16.7 | 11.5 |
| CodePlexity (Ours) | 26.7 | 21.3 | 21.0 | 15.8 | **14.1** | **25.6** | **26.6** | **24.9** | **17.3** |

*Table 1.* Comparison of question complexity metrics using mPEG on the validation set of NExT-QA. BERT and CodePlexity are trained using the outputs of the first four models (labeled as *Train* in the table), and evaluated on the rest. Text-based metrics (above) perform worse than the code-based ones (below), and our approach demonstrates the highest correlation with the models' performance.

based metrics in the upper part of the table on the held-out models, we find them to perform similarly. Notably, the BERT-based model which is trained on the questions and prediction outcomes of the four models, performs better than the learning-free baselines for InternVideo and LlaVa-Video, but fails to generalize to SeViLA. This demonstrates that the space of the natural language is not structured enough to fit a robust complexity estimation model.

In contrast, code-based metrics, shown in the lower part of Table 1 demonstrate better predictive ability overall, with even the simplest Lines of Code baseline outperforming the text-based metrics in most scenarios. Cyclomatic Complexity shows top results among all non-learning-based metrics, and our proposed approach, CodePlexity, achieves significant improvements over it by learning to identify code primitives which correlate with challenging questions.

Notably, ViperGPT is the only model that executes generated programs, making it more sensitive to code complexity — reflected in its declining success rate as program intricacy increases. Yet, intriguingly, similar declines are seen in models without access to code, suggesting our metrics capture broadly challenging patterns across architectures and training regimes. Even cutting-edge models like Tarsier and LLaVa-Video struggle with questions flagged as hard by CodePlexity, despite it being trained on older models. In addition, while all code-based metrics are affected by code correctness (see Section 10.3), CodePlexity demonstrates the greatest robustness.

In the next section, we apply the techniques introduced in Section 3.3 to identify the code structures responsible for this universal challenge to model performance.

### 4.3. Subtree Analysis

This brings us to the final aspect of our analysis: understanding these structural elements of the code that contribute to question complexity. We follow the approach proposed in Section 3.3 and identify the subtrees which are statistically correlated with a decrease in the performance $\mathcal{S}^*_{m_j}$ for three models out of out training set $\mathcal{M}_{tr}$: SeViLA,

ViperGPT, and VIOLET. In Figure 4 (right) we visualize the intersections between these three individual sets $\mathcal{S}^*$ (Equation 7) as a Venn diagram. A perceptible common trend is apparent: different architectures have their own weaknesses, but the commonalities are surprisingly frequent. We manually inspect the eight subroutines that are shared among all three sets and identify that they represent two clear patterns (subtrees are listed in Section 9.3).

The first group of primitives, manifesting in such structures as those containing *For loops* with complex control flow in them, captures reasoning about not just the content of the frame, but also its placement in a sequence of events. We provide an example of a corresponding subtree together with a question that requires this reasoning pattern in Figure 4 (left). The second group contains primitives that represent detailed analysis of specific elements (objects, relationships) within a scene. The examples include questions that require identifying the precise placement of an object within a frame.

In summary, we discovered that VideoQA methods struggle with fine-grained temporal reasoning and lack spatio-temporal, object-centric representations. This is in accord with prior studies (Buch et al., 2022; Huang et al., 2018; Liu et al., 2021) that demonstrated that naive, single-frame baselines can achieve top performance on mainstream VideoQA benchmarks, which were used to develop these methods.

Note that the granularity of the functions provided to the code generation model controls the specificity of the insights produced by our analysis tool. If too coarse — for instance, collapsing all questions into a single function — it could lose the ability to differentiate between questions. In contrast, using finer-grained functions enables more nuanced analysis, such as revealing if models tend to struggle more with "why" questions than with "how" questions.

Next, we show how our approach can be used to automatically generate a new benchmark that challenges existing approaches.

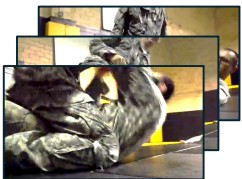
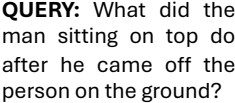
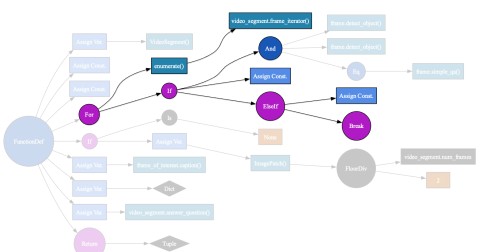
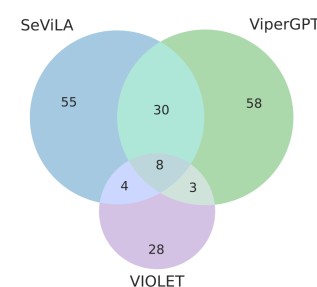

**QUERY:** What did the man sitting on top do after he came off the person on the ground?

*Figure 4.* Detailed analysis of subtrees that correlate with challenging questions among several models. We find that, although each model has its own error-modes, 8 subroutines are shared among all 3 of them (right). One of the patterns we find then analyzing the shared code structures is reasoning about the order of events (left).

| Dataset | LLaVa-Video | Tarsier | SeViLA ZS | ViperGPT | InternVideo | VIOLET | Random |
|---|---|---|---|---|---|---|---|
| NExT-QA | 82.5% | 70.9% | 64.2% | 60.0% | 50.9% | 37.7% | 20.0% |
| ATP-Hard | 77.6% | 59.8% | 54.9% | 51.8% | 24.6% | 25.4% | 20.0% |
| CodeplexQA | 65.0% | 52.5% | 43.7% | 45.8% | 29.9% | 27.6% | 20.0% |

*Table 2.* Difference in prediction accuracy between the manually annotated NExT-QA, its adversarially selected subset ATP-Hard and our automatically generated CodePlex-QA for a representative set of zero-shot VideoQA models. Our benchmark is empirically 1.9 times harder than NExT-QA, validating the effectiveness of our complexity estimation approach.

## 5. Dataset Generation

In this section, we apply our method to automatically create a new, challenging VideoQA benchmark, CodePlex-QA. We begin by detailing the source datasets and key implementation details in Section 5.1. We then compare the performance of recent VideoQA methods on the popular NExT-QA (Xiao et al., 2021) to that on CodePlex-QA in Section 5.2 to validate the effectiveness of our approach. Our dataset will be released.

### 5.1. Experimental Setup

**Source datasets.** We generate questions using 3 different datasets, all of which provide scene-graphs annotations: MOMA (Luo et al., 2021; 2022); ActivityNet (Caba Heilbron et al., 2015), which we combine with ActivityNet-Entities (Zhou et al., 2019) and ActivityNet-Captions (Krishna et al., 2017), and the ActionGenome (Ji et al., 2020) annotations for Charades (Sigurdsson et al., 2016). This results in pool of 4191 videos that are passed to our algorithm. Additionally, Section 10.2 of the appendix includes an ablation experiment that controls for video content while keeping the question generation pipeline unchanged. To this end, we use only videos from VidOR (Shang et al., 2019), matching those used to construct NExT-QA.

**Implementation details.** We use GPT-4 (OpenAI, 2023b) to generate question and answer candidates following prior work by (Mangalam et al., 2023) and leverage a state-of-the-art image captioning model, LLaVA 1.5 (Liu et al., 2023a;b), to list visual attributes of the main actors and ob-

jects in the videos. Following (Xiao et al., 2021) we generate 5 answer candidates for each question (1 correct answer and 4 distractors), and use accuracy as the evaluation metric. We set the $\delta$ in Equation 8 to select the top 10% of the data according to the estimated complexity (calibrated on NExT-QA). Further details are provided in Section 8.

### 5.2. Results

To construct CodePlex-QA, we run the generation pipeline described in Section 3.4, obtaining 20791 candidate questions (several question candidates are generated for each video). Then we calculate each question's complexity score using CodePlexity to only retain questions that meet or exceed the minimum complexity threshold as in Equation 8. The resulting datasets consists of 2261 questions. The final manual filtering to ensure the answerability of the generated questions removes only 12% of the candidates, leaving 1981 samples, all of which are used for evaluation.

We then evaluate the zero-shot baselines from our pool of methods on CodePlex-QA and report their accuracy in Table 2. We also report the results on the popular NExT-QA benchmark and on ATP-Hard (Buch et al., 2022), an adversarial split of NExT-QA, for reference. Note that the accuracy of the random baseline is the same for all the benchmarks, so the numbers are directly comparable. We observe that that the accuracy on our generated questions is significantly lower than on NExT-QA. Specifically, CodePlex-QA is 1.9 times harder than the manually annotated NExT-QA (complexity estimated by averaging the accuracy of the methods and subtracting random chance).

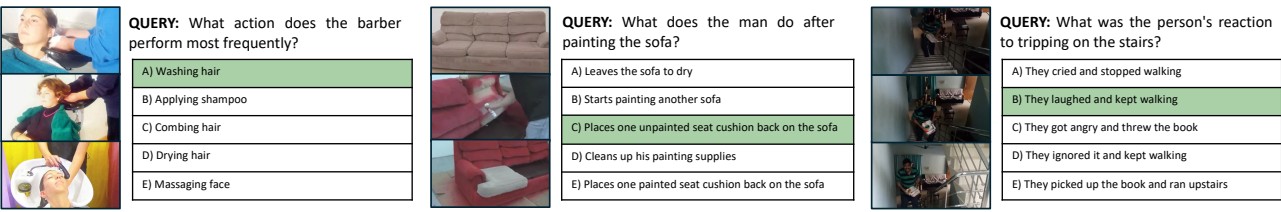

*Figure 5.* Example questions in CodePlex-QA generated with our approach. It features many challenges that are under-represented in existing, manually-designed benchmarks, motivating development of new approaches with enhanced spatio-temporal modeling capacity.

We further report models' performance on ATP-Hard, a subset of NExT-QA created using *ground truth* labels and specifically designed to be adversarial to CLIP-based models. Despite this oracle nature of ATP-Hard, CodeplexQA is substantially more challenging for the top performing models and approximately equally hard for the weakest VI-OLET baseline. ATP-Hard is somewhat more challenging than CodeplexQA for InternVideo, because it is finetuned from CLIP and the samples in ATP-Hard specifically target CLIP-based models. These results support the effectiveness of both our complexity metric and of our automatic approach for generating VideoQA benchmarks.

Figure 5 includes a representative sample of generated questions that illustrate the variety of scenarios in CodePlex-QA. They include questions that require fine-grained temporal reasoning (e.g., comparing the frequency of different actions), sequential event understanding (e.g. identifying actions that follow specific trigger events), as well as reasoning about objects. Data-driven nature of our approach ensures that CodePlex-QA highlights under-represented challenges in video-understanding. More examples are shown in the video.

Further experiments that isolate and evaluate specific components of our pipeline can be found in the appendix. In particular, in Section 10.1 we validate our question selection algorithm. Similarly, in Section 10.2 we isolate the impact of video source on dataset difficulty, confirming the effectiveness of our question generation approach.

## 6. Conclusion

We demonstrated that generated code complexity is an effective measure of question complexity in VideoQA, introducing a novel metric that outperforms existing ones. Our approach identifies subroutines associated with difficult questions across a wide range of models, providing insights into key challenges in VideoQA. Finally, we have shown how our metric can be used to automatically generate a novel benchmark – CodePlex-QA, which is 1.9 times harder for existing models than the manually labeled NExT-QA. As new methods are developed, our approach can be re-applied, ensuring continued progress in the field.

## Impact Statement

In our study, we utilize many distinct pre-trained models, each with its inherent biases, to identify challenging questions within an existing dataset. Although they have different pre-training schemes, these models likely encode similar implicit biases, owing to their training on internet scale data collections. In particular, several of our selected models, along with our visual descriptors extractor, rely on CLIP (Radford et al., 2021) as a visual encoder, meaning they likely replicate the same biases including those identified in previous studies (Agarwal et al., 2021).

Furthermore, the dataset we base the majority of our analysis on, NExT-QA (Xiao et al., 2021), is not fully representative of real-world diversity and complexity. This limitation in addition to the biases present in the chosen models can lead to skewed or incomplete analysis. In addition, our analysis' focus on interplay between the constituent syntactic elements in code may overlooks critical sources of complexity not apparent in the code structure. These include, for example, differences related to gender and ethnicity, which are not explicitly manifested in the code.

Our own proposed benchmark, CodePlex-QA, builds upon the existing datasets MOMA (Luo et al., 2021; 2022), ActivityNet (Caba Heilbron et al., 2015), and Action Genome (Ji et al., 2020), and therefore includes the same biases. We urge researchers and practitioners to refer to the relevant dataset cards. Finally, our selection methodology for filtering videos and questions may inadvertently introduce new biases, or amplify existing ones.

**Acknowledgments.** This work is in part supported by the Toyota Research Institute (TRI), AFOSR YIP FA9550-23-1-0127, ONR N00014-23-1-2355, and ONR YIP N00014-24-1-2117.

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

# Appendix

This appendix includes further details, results and discussions that were not included in the main paper due to space limitations:

1. Section 7 provides **additional technical details** for our baselines and complexity estimation methods.

2. Section 8 compliments Section 5 in the main paper providing **extra technical specifics** regarding the dataset generation pipeline.

3. Section 9 reports **additional results and analysis** to those in Sections 4.2 and 4.3 in main paper.

4. Finally, Section 10 includes important **ablations** to validate the importance of individual components of our model and data generation pipeline.

We also include a separate video with qualitative examples of the analyzed questions and samples from our new dataset at youtu.be/lMvZNpZP1WQ. Finally, we release code, models, and other materials at ceyza-guirre4.github.io/codeplexity.

## 7. Additional Technical Details

### 7.1. Merging Duplicate Subtrees

To avoid duplicated subtrees and reduce redundancy, we merge subtrees that always co-occur when one is a descendant of the other. Specifically, a subtree $S_1$ is said to always co-occur with another subtree $S_2$ if every occurrence of $S_2$ in the dataset $\mathcal{D}$ is also an occurrence of $S_1$. In such cases, since $S_2$ is always contained within $S_1$, we can merge $S_2$ into $S_1$ without losing any unique patterns.

Merging these subtrees does not risk missing important patterns because any syntactic or semantic information captured by $S_2$ is inherently included in $S_1$. This is due to the fact that $S_1$ encompasses all occurrences of $S_2$, ensuring that the features associated with $S_2$ are preserved within $S_1$. By eliminating redundant subtrees, we streamline the dataset, which can improve computational efficiency without compromising data integrity.

The merged set of subtrees $\mathcal{S}_{\text{merged}}(\mathcal{D})$ is defined as:

$$
\begin{aligned}
\mathcal{S}_{\text{merged}}(\mathcal{D}) = \mathcal{S}(\mathcal{D}) \setminus \Big\{ S_2 \in \mathcal{S}(\mathcal{D}) \,\Big|\, \exists S_1 \in \mathcal{S}(\mathcal{D}) : \\
\big( \forall T \in \mathcal{D}, \text{ISO}(T, S_2) \to \text{ISO}(T, S_1) \big) \\
\wedge \big( S_2 \subseteq S_1 \big) \Big\}.
\end{aligned}
$$

(10)

Here, $ISO(T, S)$ indicates that subtree $S$ is isomorphic to a subtree within program $T$, and $S_2 \subseteq S_1$ denotes that $S_2$ is contained within $S_1$.

By applying this merging strategy, we ensure that all significant patterns are retained. The one-hot encodings of $S_1$ and $S_2$ are identical across all programs where they appear, so merging them does not alter the representation of the data. This approach maintains the richness of the syntactic structures while optimizing the dataset for analysis.

### 7.2. Human Annotation Interface and Processing

For our human baseline we conduct an annotation effort on a subset of 150 questions from the validation set of the NExT-QA dataset. To this end, we recruited 65 human subjects via the Prolific platform (Palan & Schitter, 2018), using the provided filters to select for annotators that are proficient in English.

The annotators were shown 50 sets of 3 questions (one set at a time), where they were asked to sort the questions according to their perceived complexity by indicating which questions were the *easiest* and *hardest*. An example set and the annotation interface is shown in Figure 6. Consistency was validated by repeating pairs of questions multiple times (the third question can vary). We check that relative orders remain consistent and don't consider subjects who demonstrated low consistency. We further filter out annotations that were done in too little time, and annotators who finished the complete study in less than a minimum reasonable time. The annotations from the remaining 30 subjects were used to calculate the total ordering of the questions.

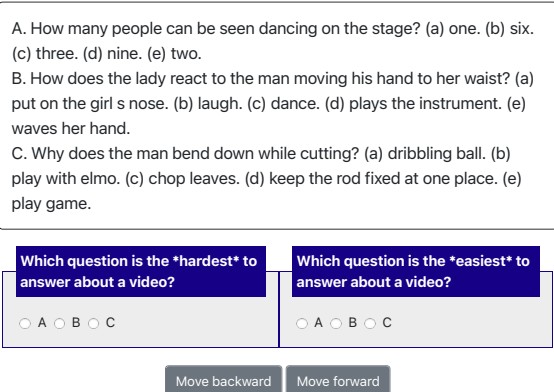

*Figure 6.* We ask human annotators to provide the relative ordering of three provided questions according to the estimated complexity of answering the question about an *unseen* video.

We compute the final order of the questions using Elo scores (Elo, 1967). Originally developed to rank chess players, the Elo system models the outcome probability of unseen comparison between a pair of entities (*eg.* chess

players or, in our case, questions) as a function of their score ratings $s_i$. In a comparison each entity's comparative performance is assumed to be Normally distributed around their score with fixed variance $\beta^2$. The probability of a favorable outcome for entity $i$ when compared to an opponent entity $j$ is given by the probability that the performance of $i$ surpasses the performance of $j$:

$$\Phi\left(\frac{s_i - s_j}{\sqrt{2}\beta}\right), \qquad (11)$$

where $\Phi$ represents the cumulative distribution of a zero-mean, unit-variance Gaussian. The scores are updated after every comparison according to the Elo update rule.

### 7.3. GPT question complexity scoring

In order to automatically estimate the complexity of a question directly from it's text without biasing the method towards any specific definition of complexity we refer to the Natural Language Processing literature which has shown that assessments made by Large Language Models correlate with human judgement (Madaan et al., 2023; Fu et al., 2023; Chiang et al., 2023; Rafailov et al., 2023). To this end, we leverage GPT-4 (OpenAI, 2023b) to generate a complexity score on a Likert scale (Likert, 1932) (ranging from one to five). We set the temperature to zero (for replicable results) and generate a single token with the score. We prompt the model as follows:

**Prompt**

> [SYSTEM] You are an assistant that -for the provided question and its corresponding answer options- estimates the complexity of answering said question about an unknown video. Return your answer as score from 1 to 5 (1 being the easiest and 5 being the hardest).
>
> [USER] I'll provide a question and its candidate answers. Estimate the complexity of answering the question about a (unseen) video. Output should ONLY be the integer score (1-5) that you assign to the question (ie. no JSON, no text, no markdown, no nothing).
>
> $query$ A: $answers[0]$, B: $answers[1]$, C: $answers[2]$, D: $answers[3]$, E: $answers[4]$

### 7.4. CodeGen details

We use the same API as in the original ViperGPT paper (Surís et al., 2023). For NExT-QA analysis we use the programs and predictions from (Surís et al., 2023), which were generated by Codex (Chen et al., 2021), a Code Completion LLM. Since Codex is no longer avaialble, for the CodePlex-QA code generations we instead use a text vari-

ant of GPT-3.5 with support for 16k context window (necessary because of the long API Specification). We prompt this model with both the API and a System message to make sure the output is usable as code. Additionally, we process the output to extract the code and format it such that it can be executed and analyzed. We include the prompt used:

**Prompt**

> [SYSTEM]Only use the functions you have been provided with."
> [SYSTEM]Only complete the code. Don't include markdown syntax (eg. ticks).
> [USER]<API Spec.>
> # $query$
> # [$answers[0]$, $answers[1]$, $answers[2]$, $answers[3]$, $answers[4]$]
> def execute_command(video, possible_answers, question):
> # Reason every step

### 7.5. Logistic Regression Parameters

We use the SciKit-Learn (Kramer & Kramer, 2016) implementation of Logistic Regression model. Each question-subtree pair has multiple labels (one per model in the training set) so we average them into a single soft label. The logistic regression model is then trained on these, allowing it to capture consensus across the different models We train until convergence and choose parameters based on the reported mean accuracy over 5 folds. The resulting model uses L2 regularization with weight $c = 1.0$ and is trained with the L-BFGS solver (Byrd et al., 1995).

### 7.6. Cyclomatic Complexity Calculation

We compute of Cyclomatic Complexity via an open source implementation[1].

## 8. Additional Question Generation Details

As noted in Section 5 of the main paper, we leverage existing video datasets with scene-graph annotations MOMA (Luo et al., 2021; 2022), ActivityNet (Caba Heilbron et al., 2015), and Action Genome (Ji et al., 2020). As necessary step we need to translate the annotations into a textual format such that a generative language model could use it. We begin by identifying the main activity and its sub-activities, including the start and end times of each. This temporal framework serves as a scaffold for the detailed enumeration of the actors and objects involved. Actors and objects are cataloged not just by their presence, but also in relation to specific sub-activities. When a high level textual

---

[1] https://radon.readthedocs.org

description is not available we leverage captioning models to generate visual descriptors of the actors in the video. More details in Section 8.1.

The resulting dataset has an average of 2.40 questions per video. The duration of each video ranges from approximately 3 seconds to 10 minutes with an average video duration of about 1.5 minutes. This diverse range of video lengths is desirable as it is conducent to generating a wide variety of questions.

The following sections describe the methods and prompts we used to translate the graphs into textual *scripts* for each specific dataset: MOMA (Luo et al., 2021; 2022) in Section 8.2, ActivityNet (Caba Heilprin et al., 2015) in Section 8.3, and Action Genome (Ji et al., 2020) in Section 8.4. Finally, Section 8.5 describes how the generated *scripts* are used to generate new questions.

## 8.1. Visual Descriptors extraction

A limitation of using Scene Graphs is that they tipically don't include visual descriptions of the nodes they relate. This is in juxtaposition with the way humans typically refer to actors and objects. To this end we describe the main actors in the video using a Captioning model. In particular, we use Llava1.5 (Liu et al., 2023a;b) to describe a single instance of the actor in the video. We leverage the included bounding box annotations for the actors in each annotated interaction. We choose the bounding box with the largest area (in pixels) in the first subactivity the actor appears in. We then crop the relevant area and zero-pad the borders to make the final image square, as this is the format that Llava1.5 was trained with. The resulting cropped image is passed into the captioning model along with a prompt modified from Llava.

**Prompt**

> "A chat between a curious human and an artificial intelligence assistant. The assistant gives helpful, detailed, and polite answers to the human's questions.
> [USER]<image>
> Look at the picture and tell me only about the person's looks that don't change. Like what they're wearing, their hair color and style. **Don't talk about where they are OR what they're doing**. (Tag is <actor_classname>).
> [ASSISTANT]

Importantly, we also pass in the textual identifier for the actor, indicated in the prompt as <actor_classname>.

## 8.2. MOMA

A significant limitation specific to MOMA is the inconsistency in annotated identifiers for object or actor through-

out the entire video. Consequently, we exclude videos in the dataset for which objects or actors cannot be reliably identified. Our implementation of the filtering process eliminates any videos from the dataset where the identities of actors are not consistently recognizable based on their $class\_name$ identifier. In practice, we define a Python function to detect 'collisions' - instances where the same identifier is used for different class names within a subactivity, or across different subastivities without a consistent mapping.

As noted in Section 5 of the main paper, we need to translate the contents of the scene-graph-in-time annotated in MOMA into a textual format such that a generative language model could use it. We now describe the method we used to translate the graphs into textual *scripts*.

We begin by identifying the main activity and its subactivities, including the start and end times of each. This temporal framework serves as a scaffold for the detailed enumeration of the actors and objects involved. Actors and objects are cataloged not just by their presence, but also in relation to specific sub-activities. We identify their class names and descriptive attributes along with arrival their departure times within each sub-activity and store these for later. We also track state changes and action, both transitive and intransitive, that occur during the sub-activities, along with the identifiers that map to the actors and objects involved.

The final script is structured in a hierarchical format, starting with the main activity title and its timeframe, followed by detailed sections for each sub-activity. These sections enumerate the actors present, and a chronological account of events, actions, and state changes. We also generate a descriptive caption for each actor involved following Section 8.1 and include it in the prompt. An example of an activity and its first sub-activity is shown:

> # **Activity:** "Dining" (0-597)
> All actors:
>
> - **ROLE:** customer. **Visual description:** The person in the picture is a woman with long, dark hair. She is wearing a white shirt and a black tie.
>
> - **ROLE:** waiter. **Visual description:** The waiter in the picture is a young man wearing a white shirt and a black tie.
>
> ## **Sub activity (0-10):** The waiter is talking to the customer or helping them into their seat
>
> - **Actors present:** customer, waiter

- **Happened during sub-activity:**
  - (attribute) waiter standing
  - (transitive action) waiter talking to customer
  - (intransitive action) waiter bending

## 8.3. ActivityNet

Although the original ActivityNet (Caba Heilbron et al., 2015) dataset didn't include scene-graphs, follow up work ActivityNet-Entities (Zhou et al., 2019) provides additional annotations for objects, attributes, relationships and actions. Further, we also use the per-subactivity captions in the ActivityNet-Captions (Krishna et al., 2017) dataset.

As was the case with MOMA, we translate the contents of the scene-graph-in-time annotated into a textual format such that it can be parsed by a generative language model. We once again divide a video into a main activity and its component subactivities, and take note of their start and end times. Actors present in a particular subactivity are listed within the subactivity description, along with their provided visual descriptions when available in ActivityNet-Entities (Zhou et al., 2019). Finally, we filter relationships such that we only keep those that involve actors and list those for each actor. An example of an activity and a subactivity is shown below:

---

# **Activity:** "doing archery" (time: 11-177)

## **Sub activity** (15-39): He loads an arrow in the bow.
All actors descriptions from subactivity:

- Visual description (time: 31): attributeclass: person - age&sex: man - hairstyle: straight - hairlength: short - haircolor: ['black'] - accessory: ['glove'] - skincolor: white - upperclothestype: t-shirt - upperclothescolor: ['white']- lowerclothestype: shorts - lowerclothescolor: ['black'] - status: ['standing', 'shooting'] - location: outdoors

- Relations for actor in subactivity:
  - person pulling bow
  - person holding arrow

---

## 8.4. Action Genome

We leverage the scene-graph annotations in ActionGenome (Ji et al., 2020) for the Charades video dataset (Sigurdsson et al., 2016) and use the annotated activity along with its duration and high level description. When available we also include the location and other descriptions. We then list the annotated actions (along with their respective start and end times). Finally, we iterate over the annotated-per-frame object-actor relationships and track when their state changes. We provide the timestamp at which the state-change occurred and the change itself to the generation model. As was the case with ActivityNet, we don't need to generate visual attributes as the high level description often provides them. An example *script* is shown below:

---

# **Activity** (duration: 30.62): "A person sits at a desk in the living room. The person laughs as they pick up a bag of groceries from under the desk."

Location: "Bedroom"

Other descriptions:

- A person is sitting at adesk they pick up a bag and then they get up

- The person is sitting at the computer desk and bends over to pick up the garbage, which he sits on his lap, and then gets up carrying the garbage.

Actions:

- Taking a bag from somewhere (9.00, 16.40)

- Sitting at a table (0.00, 29.30)

- Someone is standing up from somewhere (25.00, 30.80)

- Holding a bag (12.70, 32.00)

- Someone is laughing (0.00, 30.80)

- Sitting in a chair (0.00, 32.00)

Relation Changes (wrt. actor):

- bag goes from "'holding'" to "'touching' at 18.0"

- bag goes from "'touching'" to "'holding' at 20.0"

- bag goes from "'holding'" to "'touching' at 26.0"

- bag goes from "'touching'" to "'holding' at 27.0"

---

- chair goes from "'sitting_on'" to "None at 28.0"

- chair goes from "None" to "'not_contacting' at 28.0"

## 8.5. Question Generation from Scripts

Each textual *script* generated above is combined with a prompt requesting that the language model output *interesting* questions about the video, without specifying that they should be hard, or how *interesting* should be interpreted. The complete prompt is used to condition a Large Language Model to generate the requested questions and answer candidates in a JSON format. The chosen language model is GPT-4. We set the sampling temperature to zero and decode greedily (for replicability). We include the exact prompt used here:

**Prompt**

[SYSTEM]You generate *interesting* questions to ask about the video for which the description is provided. Pretend you don't get the exact description (ie. no exact times or player ids) but you did watch the video, so you have a notion of what happens, and when.
[SYSTEM]Return a list of Multiple Choice questions formated as a json with q, ans, dist1, dist2, dist3, dist4 keys. 'distN' are 4 distractors..
[USER]What are interesting questions to ask about this video? (description provided)
Return a numbered list of Multiple Choice questions formated as a json with q, ans, dist1, dist2, dist3, dist4 keys. 'distN' are 4 distractors
Try to use visual descriptions of the actors instead of their role sometimes (eg. the person with the red shirt instead of the waiter).

**NEVER** say subactivities. Eg. don't say "first subactivity", instead say "while the waiters served the drinks".

Video description:

—

<DESC>

—

Remember, **NEVER** say subactivities. Eg. don't say "first subactivity", instead say "while the waiters served the drinks".

When generating questions for videos from the MOMA dataset we also include an additional instruction to make sure the model doesn't refer to actors as *unclassified* when the annotations are missing descriptions:

Also avoid saying "unclassified ..." to refer to actors. If you weren't provided with a role use visual descriptions instead.

## 9. Additional Results and Analysis

### 9.1. Relation to Video Complexity

Our approach to estimating the complexity of VideoQA tasks is grounded in insights from classical complexity theory, specifically the Chain Rule for Kolmogorov Complexity. According to this rule, the complexity of a composite entity, such as a (video, question) pair, can be expressed as the sum of the complexity of one component (e.g., the question) and the conditional complexity of the other component (e.g., the video conditioned on the question), plus a logarithmic term:

$$K(x, y) = K(x) + K(y|x) + O(\log(K(x, y))) \quad (12)$$

In cases where there is minimal shared information between the video and the question, the complexity of the pair can be approximated as the sum of their individual complexities, up to this logarithmic term:

$$K(x, y) \approx K(x) + K(y) + O(\log(K(x, y))) \quad (13)$$

In this work, we focus on estimating the complexity of the question, which parallels the structure suggested by the Chain Rule for Kolmogorov Complexity. If we treat the video as $x$ and the question as $y$, then the complexity of the (video, question) pair can be thought of in a similar way, where the complexity of the question $K(y)$ is a key component of the overall task complexity. While we do not directly compute Kolmogorov complexities, this analogy provides a theoretical motivation for focusing only on the question complexity. Estimating the complexity of the question is valuable because it can later be combined with robust methods for assessing the complexity of the video to achieve a more comprehensive measure of the total task complexity in future work.

The difference between the predicted complexity of the question and the actual model performance on a given question can be used as a proxy for estimating the video's complexity. Prior work by (Wei et al., 2016) has approximated image complexity by the number of objects present in an image. We extend this approach to video complexity, utilizing the VidOR dataset (Shang et al., 2019), which provides annotations of entities and their relations in videos. Conveniently, VidOR and NextQA share the same video

source, YFCC100M (Thomee et al., 2016), allowing us to align entity counts with our models' performance on NextQA.

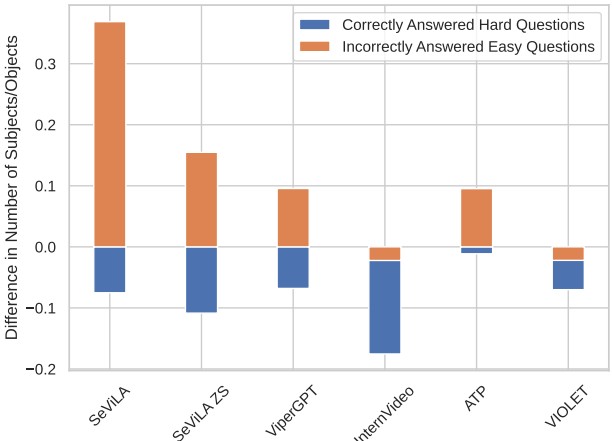

*Figure 7.* Comparison of the average number of entities (subjects and objects) in videos where models perform poorly on low complexity tasks (easier questions) versus where they perform well on high complexity tasks (harder questions). More entities make easy questions harder, while fewer entities make hard questions easier.

Figure 7 compares the difference in the average number of entities (subjects and objects) in videos where models perform poorly on low complexity tasks (easier questions) versus where they perform well on high complexity tasks (harder questions). The trend indicates that, for easier questions (low complexity), videos with more subjects/objects tend to result in poorer model performance, while for harder questions (high complexity), models perform better when fewer subjects/objects are present in the video.

This pattern supports the idea that the sources of complexity of a VideoQA task can be combined. Simple questions become more challenging when accompanied by complex videos, and difficult questions can be made easier in the presence of simpler videos. This relationship between video complexity and question complexity reinforces the importance of estimating both components. While our current approach focuses on question complexity, combining it with accurate video complexity estimations can yield a more precise measure of the overall task complexity. We leave the exploration of more accurate methods for video complexity estimation to future work.

### 9.2. Applicability to other tasks

In principle, our approach applicable to any QA domain. However, we limit the scope of this paper to VideoQA, where the reasoning complexity is significantly higher and model failure modes are more nuanced; thus, our approach has the most significant utility.

To illustrate this, we analyzed the GQA image QA benchmark (Hudson & Manning, 2019), generating programs with ViperGPT (Surís et al., 2023). For consistency, we evaluated the BLIP-2 (Li et al., 2023) VQA model (chosen as it shares the same backbone with SeViLA) and ViperGPT. We found that the overall complexity of generated programs is significantly lower in images (10% of programs with cyclomatic complexity above 5 compared to ≈45% in NExT-QA). As a result, the relationship between cyclomatic complexity and model performance is much stronger in VideoQA than in ImageQA, as evidenced by a higher coefficient of determination $R^2$ of 0.75 vs. 0.39.

Intuitively, this difference in program complexity is not surprising, as answering a question in a video typically requires analyzing more content than in a single image. In the case of NLP, the underlying tasks often involve even simpler reasoning patterns. For example, the multi-step reasoning chains in HotPotQA (Yang et al., 2018) are linear, and the dataset collected to evaluate HuggingGPT (Shen et al., 2023) has fewer than 2 module calls per prompt on average, and in simple patterns.

### 9.3. Subtrees Visualization

In this section, we present visualizations of subtrees which correlate with question that are challenging to answer for all 3 models analyzed in the main paper (see Sections 3.3 and 4.3). A majority of the nodes present in the ASTs encode non-essential information such as variable names, while we care about the actual structure and the operations being executed on the frames. For this reason, we ignore variable names and values when comparing two subtrees to one another. Similarly, we develop a tool to visualize the general structure of subtrees that performs a related node-trimming step. Finally, the visualizations of ASTs in the paper (*eg.* Figure 4 of the main paper) include an additional simplification step in which nodes are merged to aid in understanding and interpretability.

All the 8 subtrees that are shared by the 3 models in the main paper are shown in Figure 16. There are two principal patterns that can bee seen from analyzing them. The first group includes primitives that allow for temporal reasoning (Figures 8 to 13). The other common pattern group includes questions that require more detailed analysis of specific elements (objects, relationships) within a scene (Figures 14 and 15).

First, we consider the primitives necessary for temporal reasoning, *i.e.* for questions that necessitate taking into account a specific frame's placement in a sequence of events. The subtrees shown in Figures 8 and 9 both contain the control flow necessary for identifying an event that happens after a particular condition has been met. Figure 8 in particular illustrates a common pattern for finding the frame *after*

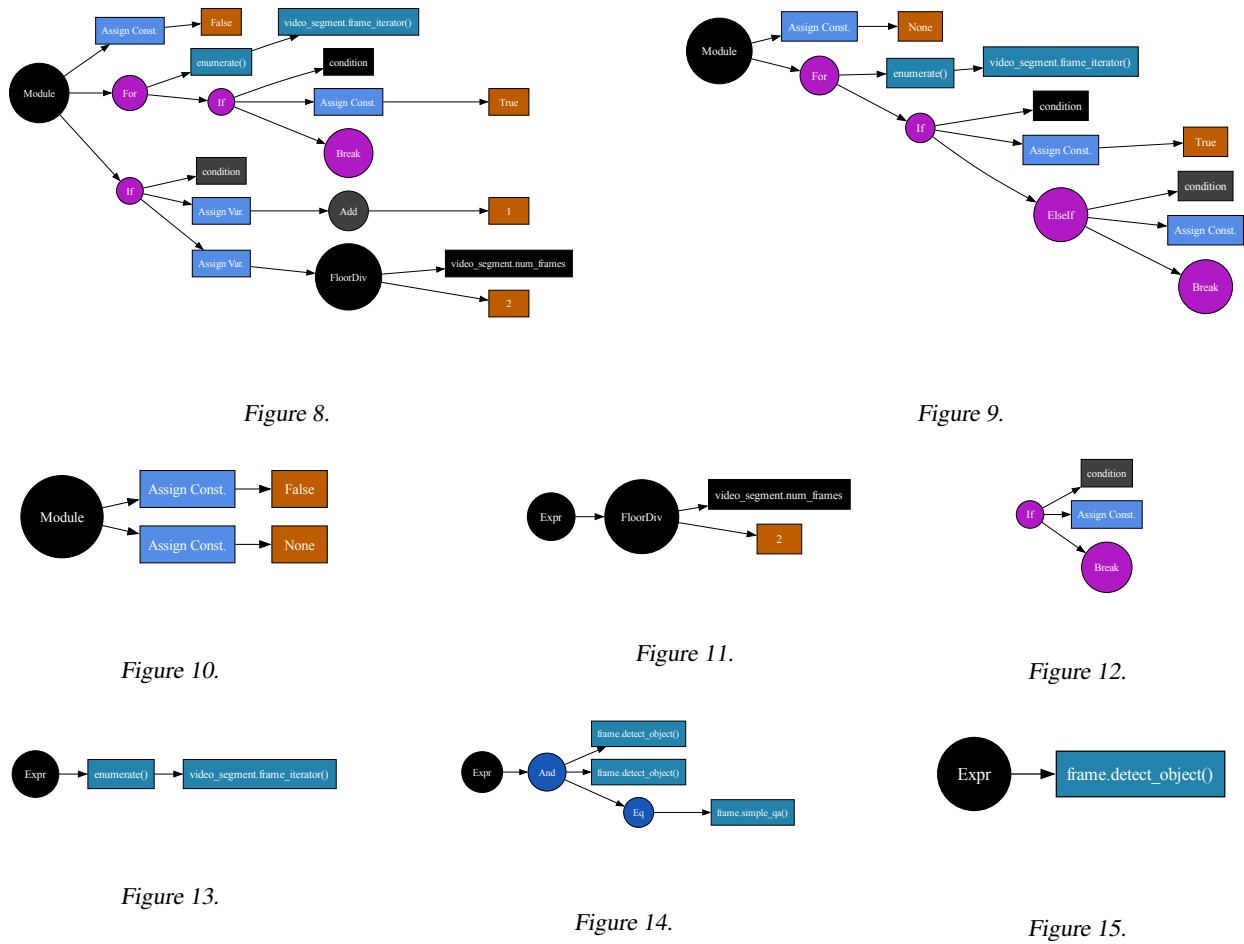

Figure 8.

Figure 9.

Figure 10.

Figure 11.

Figure 12.

Figure 13.

Figure 14.

Figure 15.

Figure 16. Subtrees identified as hard.

something has happened, with a *For Loop* that identifies the relevant part of the video, followed by an addition to look *after*. Similarly, Figure 9 is common in programs that have to identify a second condition that happens after a first one. For this reason the control-flow is slightly more complex, and includes a second conditional that is only checked for after the first condition has been satisfied.

Figures 10 to 13, while also temporal, are less obviously so. The presence of the primitives shown in Figure 10 correlates with code that includes a loop over frames in the video. This pattern is commonly used to setup for iterating over the video until a relevant frame is found. Upon identification, the first boolean variable switches to *True* and the other one is used to store the frame. Intuitively, this pattern is useful for answering questions about a specific moment of a video. The primitives in Figure 11 show code that selects a frame from the middle of the video. In practice, programs include this code as a fail-safe when searching for a specific frame, falling back to selecting the middle frame in case no satisfying frame is found (*eg.* Fig. 8). Figure 12 shows a *break* statement that will halt an iteration over the

video when a frame that meets the required criteria is seen. Intuitively, this pattern allows for the identification of the first event in the video that meets some criteria, as the break in the loop avoids overwriting the variable with frames that come in the future. And Figure 13 shows an iterator over the video, which is the main primitive necessary to consider frames in order. This primitive is often used in conjunction with others shown, *eg.* with the *break* statement in Fig. 12 to find the first frame that meets the condition.

The other common pattern group involves questions that require a more granular consideration of specific elements (objects, relationships) within a scene. For example, the subtree shown in Figure 15 is included in questions that require focusing on a single specific object or actor in a frame of the video. Relatedly, programs that include the subtree in Figure 14 require identifying at least two objects or actors and then relating them (by calling *simple_qa()*, an image question answering module).

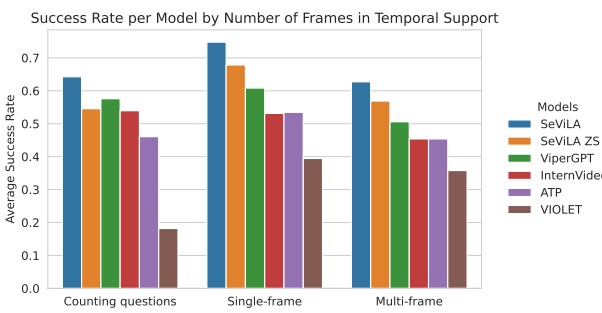

*Figure 17.* Temporal support (*i.e.* number of frames a question needs) according to the generated program. All models tested perform significantly worse on questions that require more frames. Counting questions are listed separately, as they potentially require every frame to be checked.

### 9.4. Additional Analysis Results: Temporal Support

The interpretable nature of subtrees allows us to manually identify a subset of subtrees we know correspond to subroutines that store frames. We leverage Equation 1 to count the appearance of said subtrees in each program's ASTs to find the temporal support of each question. As previous works have proposed (Mangalam et al., 2023), we validate that a significant source of question complexity in Video Question Answering is owed to the number of frames needed to answer the question (Figure 17).

### 9.5. Dataset Statistics: CodePlex-QA

The outcome of the data generation process is summarized in Table 3, which presents a breakdown of CodePlex-QA. For each source dataset, the table shows the number of questions and videos that pass the described CodePlexity filter. Importantly, a majority of the questions are associated with a unique video.

## 10. Ablations

### 10.1. Validating Question Selection Algorithm

We now validate our approach for selecting hard questions based on our complexity estimation using the NExT-QA validation set. In particular, we follow the same approach (and with the same threshold and parameters) as when constructing CodePlex-QA, and select the most challenging questions from this set. We then evaluate the same models from Section 5.2 on this subset, which we call NExT-QA$^*$, and compare to both the original NExT-QA dataset, and CodePlex-QA. This allows to separate the effects of question generation and question filtering when constructing CodePlex-QA as NExT-QA and NExT-QA$^*$ share exactly the same base set of questions.

Figure 18 indeed validates that our approach is successful in identifying a subset of NExT-QA that is more challenging for all models evaluated compared to the original dataset. However, our final dataset generation pipeline results in an even more challenging benchmark by first constructing a more diverse pool of videos and questions for the filtering approach to select from.

### 10.2. Validating Question Generation Pipeline with Same Videos

While the validation of the question selection algorithm component of our dataset construction pipeline in Section 10.1 shows that our selection algorithm is effective in identifying challenging subsets of questions, this does not fully guarantee that the higher observed complexity in CodePlex-QA in Table 2 is exclusively due to the question generation pipeline. Therefore in this section we isolate the video source effects by adapting our pipeline to use the same videos from VidOR (Shang et al., 2019) as NExT-QA.

To generate the questions, we use VidOR annotations and augment them with annotations from VidSTG (Zhang et al., 2020) and captions generated by ChatGPT (Zhang et al., 2024). We then apply our full pipeline without modifications to obtain CodePlex-QA-VidOR. Finally, we evaluate the baseline models from Table 2 and report the results in Table 4.

As Table 4 shows, CodePlexQA-VidOR remains consistently more difficult than NExT-QA across all models, despite using identical video sources. CodePlexQA-Vidor is slightly easier than our original CodePlexQA, both because of divergent data sources and because the less detailed VidOR scenegraphs limit the expressivity of the generated questions. This confirms that while video source contributes to overall dataset characteristics, the key driver of difficulty is our question generation and selection methodology, not merely differences in video content.

### 10.3. Impact of Code Generation Correctness on Complexity Metrics

To investigate the relationship between code generation correctness and the alignment of complexity metrics to problem structure, we conduct an ablation study comparing cases where the generated code produces correct answers with those where it does not. Note that "incorrect answers" do not necessarily imply that the code itself is invalid; rather, it may fail to produce the expected output.

The results are summarized in Table 5, which presents the correlation between the complexity metrics and the mPEG metric for various models. From Table 5, it is evident that the correlation between the mPEG metric and the various code-based complexity metrics is consistently higher for

| | #Questions | # Videos |
|---|---|---|
| Action Genome (Ji et al., 2020) | 1572 | 1154 |
| ActivityNet (Caba Heilbron et al., 2015) | 749 | 594 |
| MOMA (Luo et al., 2021; 2022) | 133 | 93 |

*Table 3.* Composition of CodePlex-QA in terms of number of questions and videos from each source dataset.

| Dataset | Tarsier | SeViLA ZS | ViperGPT | InternVideo | VIOLET | Random |
|---|---|---|---|---|---|---|
| NExT-QA | 70.9% | 64.2% | 60.0% | 50.9% | 37.7% | 20.0% |
| CodePlexQA | 52.5% | 43.7% | 45.8% | 29.9% | 27.6% | 20.0% |
| CodePlexQA-VidOR | 59.6% | 58.5% | 50.4% | 46.2% | 30.0% | 20.0% |

*Table 4.* Difference in prediction accuracy of zero-shot VideoQA models between the manually annotated NExT-QA, our automatically generated CodePlex-QA, and its VidOR-based variant. CodePlexQA-VidOR is consistently harder than NExT-QA despite sharing video sources, while slightly easier than the full CodePlexQA.

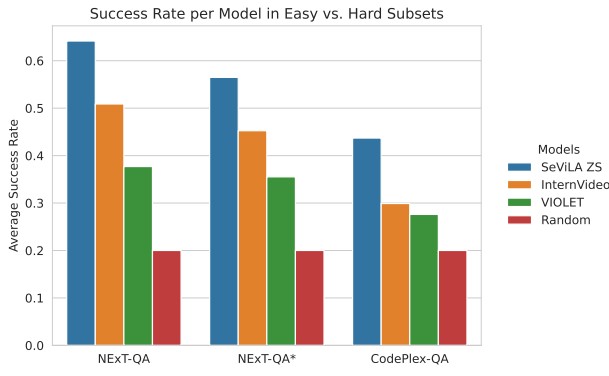

*Figure 18.* Filtering NExT-QA using our approach indeed results in a more challenging subset for all the evaluated models. However, our full dataset construction pipeline results in an even more challenging benchmark by first generating a more diverse pool of samples to select from.

cases where the generated code produces correct answers. However, we highlight that CodePlexity is a lot more robust to code generation errors than the baselines. In other words, while the correlation between models' performance and metric value improves when code is correct, CodePlexity consistently outperforms baselines in robustness to code generation errors. This underscores its practicality even when errors occur.

These results suggest that correct code generation often aligns better with problem complexity, as reflected in higher correlations with the mPEG metric. By contrast, incorrect code, while potentially valid in syntax or structure, often fails to capture the underlying complexity of the problem, thereby diluting the relationship between the metrics.

In a similar manner, Section 10.4 how that using more advanced code generation models improves the predictive power of code-based complexity metrics.

## 10.4. Impact of Code Generation Model Choice

This section evaluates the influence of the code generation model on the performance of our complexity estimation framework. Specifically, we compare ViperGPT, the primary model used in our analysis, with Recursive Visual Programming (RVP) (Ge et al., 2024), a newer model designed for visual programming tasks. Unlike traditional approaches, RVP employs a recursive code generation strategy, which systematically breaks down complex problems into manageable subproblems. This allows it to handle intricate question structures with greater flexibility.

Figure 19 illustrates the relationship between the estimated complexity of questions and the performance of Visual Programming models. For both ViperGPT and RVP, we observe a significant negative correlation between the complexity metric and the model's success rate. This trend highlights that as the estimated complexity of a question increases, the likelihood of the model correctly addressing it decreases. This correlation underscores the utility of the complexity metric as a predictive tool for identifying challenging questions.

## 10.5. Impact of Analysis Dataset Choice

To assess the generality and robustness of our findings, we replicated our analysis using a different dataset, MVBench (Li et al., 2024), which offers a diverse set of videos and questions compared to NExT-QA. Specifically, we repeat the same experimental setup as that in Section 4.1, first generating programs for the questions in MVBench, and then rerunning our pipeline to generate programs using ViperGPT, extract code based metrics, and train our CodePlexity metric. Furthermore, we additionally consider two new models in our analysis: VideoChat2 (Li

| | Train Models | | | | Validation Models | | | |
|---|---|---|---|---|---|---|---|---|
| | SeViLA | ViperGPT | ATP | VIOLET | HGA | SeViLA ZS | InternVideo | Tarsier |
| **Lines of Code** | | | | | | | | |
| Correct | 0.1373 | — | 0.1747 | 0.1455 | 0.0712 | 0.1654 | 0.2022 | 0.1475 |
| Incorrect | 0.1245 | — | 0.0540 | 0.0656 | 0.0735 | 0.0756 | 0.0831 | 0.0696 |
| **Cyclomatic Complexity** | | | | | | | | |
| Correct | 0.1702 | — | 0.2128 | 0.1930 | 0.0649 | 0.1739 | 0.2825 | 0.1634 |
| Incorrect | 0.1351 | — | 0.1118 | 0.0881 | 0.0664 | 0.0973 | 0.1388 | 0.1071 |
| **CodePlexity** | | | | | | | | |
| Correct | 0.2608 | — | 0.3128 | 0.3178 | 0.0867 | 0.2095 | 0.2877 | 0.1950 |
| Incorrect | 0.2810 | — | 0.2041 | 0.2542 | 0.1087 | 0.1839 | 0.1700 | 0.1857 |

*Table 5.* Correlation of complexity metrics with mPEG for cases where the generated code produces correct and incorrect answers.

et al., 2024) and Llava-NExT (Liu et al., 2024) as these represent the state of the art on the MVBench dataset.

We first visualize the correlation between code-based complexity metrics and the performance of various VideoQA models on MVBench in Figure 20. Consistent with our observations on NExT-QA, we found that code-based complexity metrics exhibit a strong negative correlation with model performance on MVBench. Specifically, both Lines of Code and Cyclomatic Complexity continued to demonstrate a consistent and strong correlation, indicating that questions requiring more intricate code are more challenging for all the models evaluated. This is despite the base code generation model ViperGPT (and RVP) performing worse on MVBench than on NExT-QA, where they achieve accuracies of 38.4% and 35.0%, respectively. For reference, these are comparable to recent approaches such as GPT-4V (run with 16 frames as input at 512×512 resolution), which obtains 43.5%, and VideoChat (Li et al., 2024) (also with 16 frames), which achieves 35.5%.

We further conducted a systematic evaluation of different code-based metrics using the mPEG metric on the validation set of MVBench, summarized in Table 6. Our proposed CodePlexity metric significantly outperformed the naive code complexity measures, such as Lines of Code and Cyclomatic Complexity. CodePlexity achieved higher predictive accuracy in estimating question difficulty across all evaluated models on MVBench. Note that CodePlexity generalizes to the held-out models in our analysis (VideoChat2 (Li et al., 2024) and Llava-NExT (Liu et al., 2024)).

These consistent results across two distinct datasets suggest that code-based complexity metrics, and CodePlexity in particular, are effective tools for assessing question difficulty in VideoQA tasks regardless of the dataset's characteristics.

## 10.6. Training on CodePlexQA

We split CodePlex-QA into train and val sets and fine-tuned SeViLA. We observed that training on CodePlex-QA indeed can help improve the performance of VideoQA models on these hard questions (accuracy increases from 44.8 to 47.3). However, the gap is narrower compared to fine-tuning the same method on NExT-QA (accuracy improvement from 64.2 to 73.4). This suggests that our analysis uncovered deeper reasoning limitations in existing VideoQA approaches, which may not be easily resolved by simply increasing the amount or diversity of the training data.

| | Train Models | | | Val. Models | |
|---|---|---|---|---|---|
| | InternVideo | SeViLA ZS | Tarsier | VideoChat2 | LLaVA-NeXT |
| Dependency Tree Depth | 6.5 | 6.2 | 19.7 | 16.5 | 12.2 |
| GPT-4 (OpenAI, 2023b) | 2.3 | 0.0 | 14.8 | 12.8 | 8.9 |
| BERT (Kenton & Toutanova, 2019) | 25.5 | 11.6 | 18.7 | 20.9 | 21.6 |
| Lines of Code | 4.4 | 9.5 | 12.5 | 10.3 | 9.2 |
| Cyclomatic Complexity | 13.0 | 9.6 | 6.1 | 5.2 | 4.5 |
| CodePlexity (Ours) | 41.3 | 33.0 | 44.6 | **29.9** | **27.5** |

*Table 6.* Comparison of question complexity metrics using mPEG on the validation set of MVBench. CodePlexity is trained on the first three models. Our approach demonstrates the highest correlation with the models' performance.

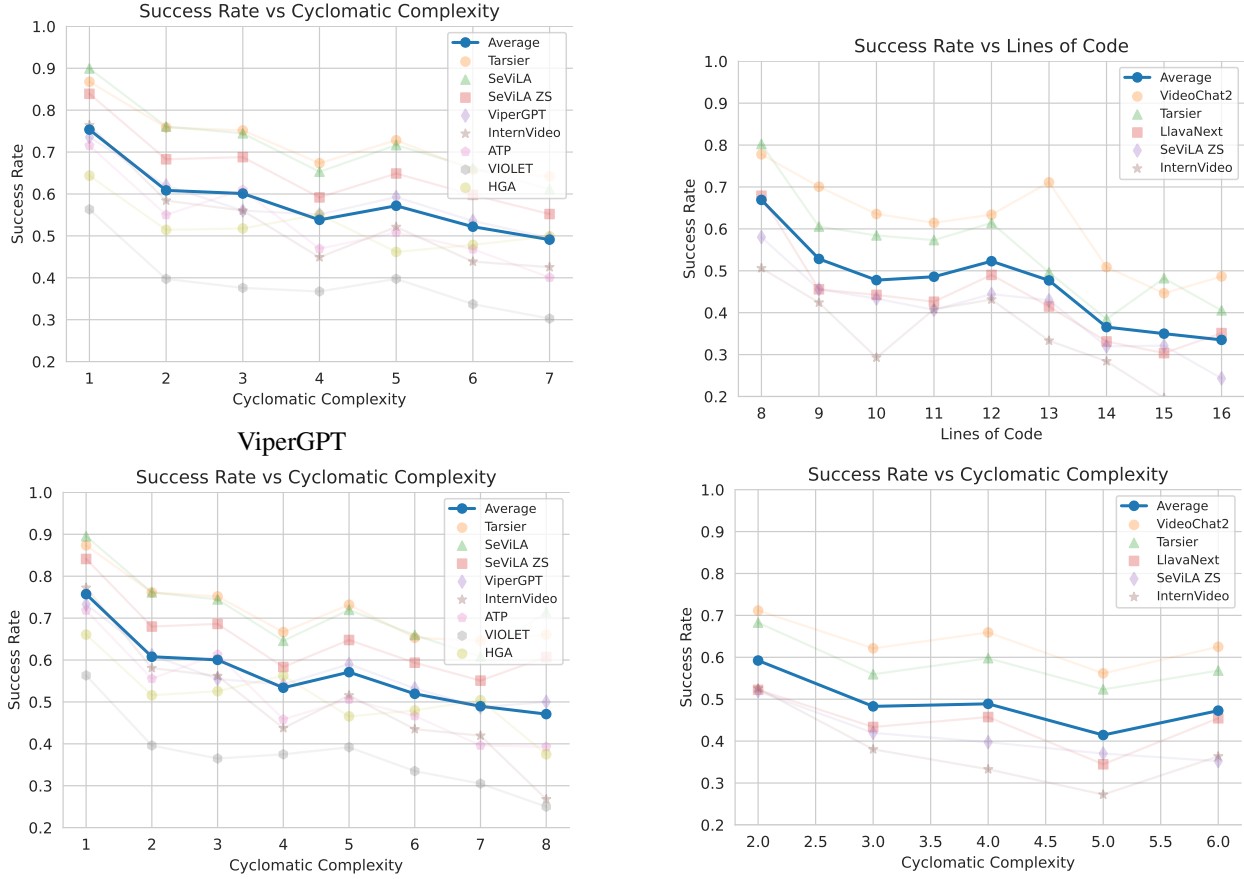

ViperGPT

RVP

*Figure 19.* Comparison of code-based complexity metrics when using different code generation models. Lines in both cases show significant negative correlation of complexity metric with model performance for both variants.

*Figure 20.* Correlation of VideoQA models' success rate on MVBench for various approaches for estimating question complexity. As was the case for NExT-QA, we observe that code complexity correlated strongly with question complexity.

