# OpenReview forum: "Understanding Complexity in VideoQA via Visual Program Generation"
_ICML.cc/2025/Conference — ICML 2025 poster_

### Official Review · Reviewer_tu7h · 2025-03-07

**Overall Recommendation:** 3

**Summary:**

This paper proposes a data-driven approach to assess question complexity in VideoQA tasks by leveraging the complexity of generated code as a proxy. The mPEG results on NExT-QA and MVbench demonstrate the effectiveness of Codeplexity. Furthermore, the newly introduced dataset, CodePlex-QA, features more challenging questions than those in NExT-QA and the ATP-hard subset, validating the feasibility of this method for creating more difficult datasets.

**Claims And Evidence:**

Yes. The claims are clearly proved.

**Essential References Not Discussed:**

No.

**Experimental Designs Or Analyses:**

Q4. The explanation of the logistic regression model training process is somewhat unclear. Specifically, the authors state that each $\left(\mathbf{x}_i, y_i^{(j)}\right)$ pair is a distinct instance (L177 Right). To clarify, does this imply that for a single question $\mathbf{x}_i$ and its valid subtrees, there are four corresponding labels $y_i^{(0)}, y_i^{(1)}, y_i^{(2)}, y_i^{(3)}$, each being either 1 or 0? If this is the case, how is the regression model trained? Please correct any inaccuracies in my understanding.

**Methods And Evaluation Criteria:**

Below are some concerns about the proposed method.

Q1. **Applicability Beyond VideoQA:** While CodePlexity is applied to analyze the complexity of VideoQA tasks, it appears to be fundamentally a language-based approach with no direct connection to video content (L042 Right). This raises the question of why the authors have limited its application to VideoQA. Could the same methodology be effectively applied to QA datasets in the NLP domain, such as HotpotQA? Clarifying its broader applicability would strengthen the method's relevance.

Q2. **Code Generation Model:** The authors use ViperGPT for code generation, which relies on function templates and few-shot examples to translate questions into code. This raises two concerns:
(i) Does the granularity of functions influence subtree generation? For instance, functions like `llm_query` and `simple_query` are partially similar. What would happen if these were merged or further decomposed into more fine-grained functions, such as `query_how` or `query_why`?
(ii) Are few-shot examples used in the prompt? If so, how do the quality and quantity of these examples impact the results? Addressing these points would provide deeper insights into the robustness of the code generation process.

Q3. **Dataset Construction and Fairness:** The CodePlex-QA dataset is constructed using videos different from those in NExT-QA, which may introduce unfairness due to variations in the inherent complexity of the videos. To enhance the robustness and credibility of the pipeline, a comprehensive ablation study could be conducted:
(i) Apply the pipeline to construct QA pairs using the same video sources as NExT-QA.
(ii) Use the same threshold to filter out a challenging subset from NExT-QA and evaluate its accuracy.
Such an approach would ensure a fairer comparison and make the pipeline more convincing.

**Other Comments Or Suggestions:**

None.

**Other Strengths And Weaknesses:**

Strengths:

1. The use of code as a proxy for assessing question complexity is novel and effective.

2. The method is supported by comprehensive theoretical proofs and statistical analysis, demonstrating its feasibility.

**Questions For Authors:**

See above.

**Relation To Broader Scientific Literature:**

This paper proposes an effective method for analyzing question complexity in QA tasks by using code as a proxy, an approach that is both reasonable and clear. Statistical analysis demonstrates that CodePlexity outperforms human evaluation and traditional complexity assessment methods. Additionally, the CodePlex-QA dataset confirms the pipeline's reliability in constructing substantially more challenging QA datasets.

**Theoretical Claims:**

Yes. I think the theoretical claims are no problems.

---

> ### Author Rebuttal · Authors · 2025-04-01
>
> Thank you for your detailed review and suggestions. We appreciate your recognition of the novelty, clarity, and effectiveness of our approach, and of the extensiveness of our theoretical and empirical analysis. We respond to your comments below.
>
> ---
>
> **Applicability beyond VideoQA.**
>
> Indeed, our approach is, in principle, applicable to any QA domain. To illustrate this, we analyzed the GQA image QA benchmark (Hudson et al., CVPR'19), generating programs with ViperGPT. For consistency, we evaluated BLIP-2 (Li et al., ICML'23) (analogous to SeViLA) and ViperGPT. We found that the overall complexity of generated programs is significantly lower in images (10% of programs with cyclomatic complexity above 5 vs.~45% in NExT-QA). As a result, the relationship between cyclomatic complexity and model performance is much stronger in VideoQA than in ImageQA  (coefficient of determination R² of 0.75 vs. 0.39).
>
> Intuitively, this difference in program complexity is not surprising, as answering a question in a video typically requires analyzing more content than in a single image. In the case of NLP, the underlying tasks often involve even simpler reasoning patterns. E.g., the multi-step reasoning chains in HotPotQA are linear, and the dataset collected to evaluate HuggingGPT (Shen et al., NeurIPS'23) has fewer than 2 module calls per prompt on average, and in simple patterns.
>
> These findings suggest that while CodePlexity is broadly applicable, its most significant utility lies in VideoQA, where the reasoning complexity is significantly higher and model failure modes are more nuanced. We will include these results, along with a discussion on broader applicability, in the manuscript.
>
> ---
>
> **Code generation model.**
>
> Indeed, the specifics of the CodeGen model used—such as module granularity and few-shot examples—affect the generated programs, which affects subtree encodings. Importantly, Section 10.3 shows our approach generalizes to different CodeGen models.
>
> **(i) Effect of function granularity:** This question is particularly insightful. In the extreme, if all questions mapped to a single VideoQA function, our complexity metrics would assign the same score to every question, losing all discriminatory power. Conversely, more granular functions enable finer-grained analysis—e.g. distinguishing whether models struggle more with “why” or “how” questions. This relates to our discussion above. We’ll add this to the paper.
>
> **(ii) Impact of few-shot examples:** We use the official ViperGPT prompts and examples. We show our approach generalizes across different visual programming models in Section 10.3, despite changes in the APIs and modules. Similarly, Section 10.2 shows CodePlexity is more robust to code gen errors than the baselines. Thus, we expect that changes in the few-shot examples would not significantly alter the overall trends.
>
> ---
>
> **Dataset construction ablation.**
>
> Thank you for this insightful suggestion. Please note that we have already provided an ablation of a part of our dataset construction pipeline in Section 10.1, where we show that applying our selection algorithm to the existing questions in NeXT-QA results in a more challenging subset. However, this does not isolate the video source effects.
>
> To address this, following your suggestion, we now adapt our pipeline to use the same videos from VidOR (Shang et al., ICMR'19) as NExT-QA. To generate the questions, we use VidOR annotations and augment them with annotations from VidSTG (Zhang et al., CVPR'20) and captions generated by ChatGPT (Zhang et al., arXiv'24). We then apply our full pipeline without modifications to obtain CodePlexQA-VidOR. Finally, we evaluate the same baseline models and report the results below.
>
> |Dataset|Tarsier|SeViLAZS|ViperGPT|InternVideo|VIOLET|Random|
> |-|-|-|-|-|-|-|
> |NExT-QA|70.9%|64.2%|60.0%|50.9%|37.7%|20.0%|
> |CodePlexQA|52.5%|43.7%|45.8%|29.9%|27.6%|20.0%|
> |CodePlexQA-VidOR|59.6%|58.5%|50.4%|46.2%|30.0%|20.0%|
>
> As the table shows, CodePlexQA-VidOR remains consistently more difficult than NExT-QA across all models, despite using identical video sources. CodePlexQA-Vidor is slightly easier than our original CodePlexQA, both because of divergent data sources and because the less detailed VidOR scenegraphs limit the expressivity of the generated questions. This confirms that while video source contributes to overall dataset characteristics, the **key driver of difficulty is our question generation and selection methodology**, not merely differences in video content.
>
> ---
>
> **Explanation of logistic regression training.**
>
> Thank you for bringing that up. Each question-subtree pair does have multiple labels (one per model in the training set), but we average them into a single soft label. The logistic regression model is then trained on these, allowing it to capture consensus across the different models. We will clarify this in the Methodology section.

---

> > ### Comment · Reviewer_tu7h · 2025-04-07
> >
> > Thanks for your reply! This really solves my doubts, and I will keep my positive score.

---

### Official Review · Reviewer_V8ky · 2025-03-14

**Overall Recommendation:** 3

**Summary:**

This paper proposes a data-driven approach to analyzing query complexity in Video Question Answering (VideoQA). They design an automatic approach that leverages recent advances in code generation for visual question answering, using the complexity of generated code as a proxy for question difficulty. They demonstrate that this measure correlates significantly better with model performance than human estimates. They construct a new benchmark that is 1.9 times harder than the popular NExT-QA.

**Claims And Evidence:**

I want to mention that I'm not convinced by the pre-assumed claim in this paper, which is that "we should define question complexity by model performance instead of by humans". I believe it is always up to humans, because models eventually learn from human. The performance difference (human think a question hard but model performance is high) can be because of lots of reasons, e.g. dataset construction and evaluation method. Specifically, human can think the "Where is this?" question in Figure1 hard because they think of many perspectives e.g. campus, city wise location, country wise location, but model many only think about restricted perspectives, e.g. whether it is outdoor or indoor. These can all be affected by the answers in the data. Therefore, I'm not convinced that question complexity is decided by models. Instead, they should be decided by humans.
If we see this paper as finding and auto-generating harder questions for certain videoQA models, then it's more appropriate.

**Essential References Not Discussed:**

N/A

**Experimental Designs Or Analyses:**

Many models are not that up to date, e.g. VIOLET, InternVideo, HGA. Why not experiment with LLaVa-Video and LLaVa-one-vision? These are specially designed for video tasks, and better than LLaVa-Next.
One small comment is to move the result table of the newer model e.g. LLaVa-next to the front.

**Methods And Evaluation Criteria:**

It's hard to interpret the result table. E.g. in Table 1, how to read the results? What does Train Models and Val Models mean? How do they relate to each other?

**Other Comments Or Suggestions:**

See above.

**Other Strengths And Weaknesses:**

I wonder if the generated  CodePlex-QA benchmark can be used as training data and improve these tested video QA models in the paper?

**Questions For Authors:**

See Claims And Evidence. Happy to raise score if questions are addressed.

**Relation To Broader Scientific Literature:**

It's related to visual programing.

**Theoretical Claims:**

N/A

---

> ### Author Rebuttal · Authors · 2025-04-01
>
> Thank you for your review and valuable feedback. We address your comments individually below.
>
> ---
>
> **On defining complexity via model performance**
>
> We appreciate your perspective that human perception plays a fundamental role in defining question difficulty. However, our work does not claim that model performance alone defines question complexity in a universal sense. Instead, it offers a systematic, adaptable tool for uncovering *common failure modes* across a diverse set of VideoQA *models*. This is especially useful for benchmarking and diagnosing model capabilities — areas where human intuition can be inconsistent or coarse-grained.
>
> Our approach does not preclude human judgment, but augments it with an automatic, scalable metric that is sensitive to the kinds of reasoning models struggle with. We strive to make this clear throughout the paper (e.g., see L026-027 in the abstract, or L077-080, L101-104 in the introduction). If there are specific instances where our framing could be improved, we would greatly appreciate any suggestions on how to refine them further.
>
> ---
>
> **Clarifying Table 1**
>
> Thank you for the detailed feedback. As described in L230-240 in Section 4.1, models are split into training and validation sets ($M_{tr}$ and $M_{val}$) to assess the generalization of complexity metrics. “Train Models” refers to models whose outputs are used to learn metrics like CodePlexity and fine-tuned BERT, while “Val Models” are held out for evaluation. This protocol ensures that the learned metrics do not overfit to specific architectures or training signals. To improve clarity, we will update the caption of Table 1 to explicitly define Train and Val Models, ensuring readers can easily interpret the results.
>
> ---
>
> **Results with more recent VideoQA models**
>
> We include models across architectural types (graphNN-based, transformer-based, codegen-based) and training paradigms (supervised, contrastive, zero-shot). Please note that Tarsier was the state-of-the-art *zero-shot* model on NExT-QA at the time of writing, and it shares architectural principles with methods like LLaVA-Video. In response to your suggestion, we have now added LLaVA-Video to our analysis and report the updated Tables 1 and 2 below. The new results further support our claims that CodePlexity is an effective question complexity metric (Table 1) and that our CodePlexQA benchmark is challenging for a wide spectrum of VideoQA models (Table 2). We will include the updated tables in the final version of the manuscript.
>
> **Table 1:** Comparison of question complexity metrics using mPEG on the validation set of NExT-QA. BERT and CodePlexity are trained on the first four models ($M_{tr}$).
> | | SeViLA | ViperGPT | ATP | VIOLET | HGA | SeViLA ZS | InternVideo | Tarsier | Llava-Video |
> |-|--------|----------|-------|--------|------|-----------|-------------|---------|---------|
> | **Type** | Train | Train | Train | Train | Val | Val | Val | Val | Val |
> | **Dependency Tree Depth**| 12.9 | 7.9 | 11.1 | 15.9 | 7.4 | 13.5 | 17.7 | 10.1 | 6.9 |
> | **GPT-4** | 9.6 | 8.9 | 11.6 | 5.8 | 7.8 | 14.6 | 13.9 | 10.8 | 5.2 |
> | **BERT** | *12.5* | *6.0* | *18.3*| *17.3* | 7.7 | 14.3 | 21.1 | 10.8 | 11.4 |
> | **Lines of Code** | 16.4 | 15.3 | 14.2 | 12.0 | 9.9 | 16.2 | 17.5 | 14.4 | 9.38 |
> | **Cyclomatic Complexity**| 18.2 | 14.2 | 18.7 | 15.9 | 8.9 | 17.2 | 24.2 | 16.7 | 11.5 |
> | **CodePlexity (Ours)** | *26.7* | *21.3* | *21.0*| *15.8* | **14.1** | **25.6** | **26.6** | **24.9** | **17.3** |
>
> **Table 2:** Difference in prediction accuracy between the manually annotated NExT-QA and our automatically generated CodePlexQA for a representative set of zero-shot VideoQA models.
> | Dataset       | LlaVa-Video | Tarsier | SeViLA ZS | ViperGPT | InternVideo | VIOLET | Random |
> |---------------|---------|---------|-----------|-------------|-------------|--------|--------|
> | **NExT-QA**       | 82.5% | 70.9%   | 64.2%     | 60.0% | 50.9%       | 37.7%  | 20.0%  |
> | **ATP-Hard**      | 77.6% | 59.8%   | 54.9%     | 51.8% | 24.6%       | 25.4%  | 20.0%  |
> | **CodeplexQA**          | 65.0% | 52.5%       | 43.7%     | 45.8% | 29.9%       | 27.6%  | 20.0%  |
>
>
> We will move the MVBench evaluation, including LLaVa-next results, to the main paper.
>
> ---
>
> **Using CodePlex-QA for training**
>
> Thank you for this insightful suggestion. To investigate this, we have split CodePlex-QA into train and val and fine-tuned SeViLA. We observed that training on CodePlex-QA indeed can help improve the performance of VideoQA models (accuracy increases from 44.8 to 47.3). However, the gap is narrower compared to fine-tuning the same method on NExT-QA (accuracy improvement from 64.2 to 73.4). This suggests that our analysis uncovered deeper reasoning limitations in existing VideoQA approaches, which may not be easily resolved by simply increasing the amount or diversity of the training data. We will include these results, along with additional discussion, in the final version of the manuscript.

---

> > ### Comment · Reviewer_V8ky · 2025-04-06
> >
> > Thanks! I’ve raised my score to 3.

---

### Official Review · Reviewer_rPct · 2025-03-14

**Overall Recommendation:** 3

**Summary:**

This work focuses on important issues in the VideoQA domain and presents a novel approach that provides new perspectives for future model evaluation and benchmark dataset construction. The core contribution is to propose a data-driven approach to systematically identify and analyze model-specific weaknesses in VideoQA tasks.

Specifically, the paper presents a visual program generation approach to analyzing the complexity of questions in video question and answer (VideoQA) tasks. The authors measure the difficulty of a question by converting a natural language question into executable code and utilizing the structural complexity of the code. Specifically, the paper proposes the CodePlexity approach, which parses the abstract syntax tree (AST) of the code, extracts the subtree structure, and trains a logistic regression model to predict how challenging different types of problems are for the model.

**Claims And Evidence:**

See weakness

**Essential References Not Discussed:**

Related works are clear

**Experimental Designs Or Analyses:**

No serious flaws found

**Methods And Evaluation Criteria:**

See weakness

**Other Comments Or Suggestions:**

None

**Other Strengths And Weaknesses:**

Strengths

This paper focuses on an important research problem in VideoQA, aiming to systematically identify and analyze model-specific challenges. The proposed approach is both methodologically sound and conceptually novel, leveraging visual program generation to assess question complexity in a unique way. Additionally, the paper is well-structured, clearly written, and effectively presents its methodology, results, and contributions.

Weaknesses

Since the estimated complexity is derived from the generated code rather than the intrinsic difficulty of the question itself, the benchmark primarily reflects model-specific difficulty rather than absolute question difficulty, which may restrict its broader applicability. Consequently, long-term impact and reliability of CodePlex-QA benchmark remain unclear, as its difficulty could be tightly coupled to the specific capabilities of the models used for generation. That being said, I acknowledge the potential of the proposed method and thus maintain a positive rating for now.

**Questions For Authors:**

The benchmark created by this method primarily reflects relative difficulty as defined by the specific code generator used as a reference point. As models rapidly evolve these days, the benchmark may quickly become outdated or lose its effectiveness in distinguishing between truly challenging questions. Given that model capabilities are difficult to quantify precisely, how can we ensure that the benchmark remains a faithful and reliable measure of VideoQA complexity as models improve? Moreover, without a stable and model-agnostic difficulty definition, how can future researchers determine whether the benchmark continues to serve as a meaningful evaluation tool?

**Relation To Broader Scientific Literature:**

This work focuses on an important research problem, and the proposed method is novel.

**Theoretical Claims:**

No serious flaws found

---

> ### Author Rebuttal · Authors · 2025-03-31
>
> Thank you for your thoughtful and encouraging review. We appreciate your recognition of the novelty, clarity, and potential impact of our approach. We respond to your questions and concerns individually below.
>
> ---
>
> **Model-specific vs intrinsic difficulty.**
>
> We acknowledge that our metric does not aim to capture an intrinsic notion of question complexity. Instead, it offers a systematic, adaptable tool for uncovering *common* failure modes across diverse VideoQA *models* (see L026-027 in the abstract, or L077-080, L101-104 in the introduction). This is a deliberate design choice. To the best of our knowledge, no universally accepted definition of "intrinsic question difficulty" in VideoQA exists, and all heuristic-based attempts to define it so far have not stood the test of time [1, 2, 3]. Rather than seeking an elusive 'intrinsic' complexity measure, our approach provides empirical insights grounded in real model behavior. This makes it a valuable *complement* to any future efforts to define question difficulty more formally.
>
> ---
>
> **Influence of the code generator on the complexity metric.**
>
> The choice of code generation model is indeed an important factor in our methodology. To investigate this, we have analyzed the robustness of code-based complexity metrics to code generation errors in Section 10.2 and re-ran our experiments using the recent RVP CodeGen approach (Ge et al., 2024) as a substitute for ViperGPT in Section 10.3. The results demonstrate that, while program correctness does affect the predictive power of code-based metrics, CodePlexity is a lot more robust to code generation errors than the baselines. Moreover, the performance of code-based metrics can benefit from advancements in visual programming models, as they can provide richer and more accurate program representations.
>
> ---
>
> **How can we ensure the benchmark remains reliable as models improve?**
>
> Ensuring benchmark relevance over time is a well-known challenge in machine learning, and we have designed our approach with this in mind. Specifically, our work does not introduce a fixed benchmark, but rather a *data-driven framework* for estimating question complexity in VideoQA. This framework enables the automatic generation of challenging benchmarks that evolve alongside advancements in model capabilities. Unlike static, human-designed datasets, our method allows the complexity metric to be recomputed and the dataset to be regenerated as new models emerge. This adaptability ensures that CodePlex-QA remains a relevant and challenging evaluation tool as the field advances.
>
> Furthermore, as demonstrated in Sections 4.2 and 10.4, our approach generalizes across datasets and model families—including recent video-language models like Tarsier and LLaVA-NeXT. This indicates that our method is not overly dependent on any single model or model family, further reinforcing its reliability as models continue to evolve.
>
> ---
>
> **Without a stable, model-agnostic definition of difficulty, how can future researchers rely on the benchmark?**
>
> We appreciate the importance of this question for ensuring meaningful evaluation over time. While our method does not define complexity in an absolute, model-agnostic sense, we argue that such a definition is not only difficult to establish but may also be fundamentally impractical. Unlike domains such as mathematics or logic, where problem complexity can be grounded in formal systems, VideoQA involves rich, ambiguous, multimodal inputs where “difficulty” is inherently contextual and model-dependent.
>
> Rather than attempting to define question complexity in the abstract, our goal is to offer a *practical*, *empirical*, and *extensible* framework for estimating it based on observed model behavior. CodePlexity is intentionally data-driven: by analyzing model failures, we can surface interpretable, reproducible signals of what these models find challenging. As models evolve, the complexity metric can be re-learned to reflect their capabilities. By continuously adapting to emerging models, our approach ensures that future researchers can still use it as a meaningful evaluation tool, precisely because it is not fixed in absolute terms.
>
> ---
>
> **References:**
>
> [1] Buch, Shyamal, et al. "Revisiting the "video" in video-language understanding." Proceedings of the IEEE/CVF conference on computer vision and pattern recognition. 2022.
>
> [2] Huang, De-An, et al. "What makes a video a video: Analyzing temporal information in video understanding models and datasets." Proceedings of the IEEE Conference on Computer Vision and Pattern Recognition. 2018.
>
> [3] Liu, Xin, et al. "No frame left behind: Full video action recognition." Proceedings of the IEEE/CVF conference on computer vision and pattern recognition. 2021.

---

> > ### Comment · Reviewer_rPct · 2025-04-03
> >
> > Thanks for the clarification. I generally agree with the author's responses and thus maintain my positive overall recommendation for this work.

---

### Official Review · Reviewer_rEFD · 2025-03-25

**Overall Recommendation:** 4

**Summary:**

## update after rebuttal

The paper introduces an interesting method of relying on visual programs to evaluate the complexity of VideoQA task. The authors develop methods to analyze the code complexity to estimate the question complexity. The proposed method, CodePlexity, correlates better with the model performance compared with human judgments. To show the utility of this approach, the authors also use this approach to create a dataset CodePlex-QA, which is empirically 1.9x more difficult than NextQA.

**Claims And Evidence:**

The evidence is overall convincing. One concern is that there might not be a gold definition of "question complexity". It is more rigid to claim that this method can find the questions that are challenging for existing video models.

**Essential References Not Discussed:**

N/A

**Experimental Designs Or Analyses:**

Overall the experiment designs are valid and the analysis is interesting. Many experiments are added in the appendix.

**Methods And Evaluation Criteria:**

The approach is novel and makes sense.

A concern is that the approach only depends on the question part of the VideoQA. Nevertheless, the visual content, for example, the length of the video, may have an important role. The approach might be better if it incorporates visual information in some way.

**Other Comments Or Suggestions:**

N/A

**Other Strengths And Weaknesses:**

Strengths:
The method is novel, interesting, and makes sense. The idea of evaluating VideoQA questions with visual programming is smart, and the authors show this method's effectiveness through their experiments. This work can be inspiring to many follow-up works on video LLMs.

Weaknesses:
The approach, including the baselines, relies only on the "text" questions but ignores the visual content. The video part may play an important role, like how long the video is, and how many entities are there in a frame. Existing methods of visual programming ignore such information and may limit the proposed methods' effectiveness.

**Questions For Authors:**

1. Is there a possibility that given the same question, its complexity can be very different given different videos? How would you address this?

2. Just curious, how well are visual program methods on NExT-QA and MVBench? From my experience, I expect their accuracy to be relatively poor.

The authors have answered the questions in rebuttal.

**Relation To Broader Scientific Literature:**

The research question is very interesting and is a focus of current research. People are thinking of ways to build better video benchmarks, and this paper provides insights into this direction.

**Theoretical Claims:**

The paper contains some deductions on the metric for code complexity, and they make sense.

---

> ### Author Rebuttal · Authors · 2025-03-31
>
> Thank you for the thoughtful and constructive feedback. We are encouraged by your positive assessment of our method's novelty and its potential impact on the VideoQA community. We respond to your questions and concerns individually below.
>
> ---
>
> **“Gold” vs model-specific question complexity**
>
> We acknowledge that defining question complexity is inherently challenging, and our approach does not aim or claim to provide a definitive measure. Instead, it offers a systematic, adaptable tool for uncovering *common* failure modes across a diverse set of VideoQA *models*. We try to be explicit about this in the paper (e.g. see L026-027 in the abstract, or L077-080, L101-104 in the introduction). If you have specific concerns about our phrasing, we would greatly appreciate any suggestions on how to make this distinction even clearer.
>
> ---
>
> **The role of the visual information**
>
> We agree that visual content can influence question difficulty. While our focus is on question-based complexity estimation, we provide both theoretical and empirical analysis of incorporating visual content in Section 9.1 of the supplement. Specifically, we show that video statistics (e.g., number of entities in a frame) correlate with residual difficulty not explained by CodePlexity. That is, as you suggested, visual features can complement our question-centric metric and serve as a promising avenue for further research. Importantly, our method achieves strong predictive performance even without incorporating visual features, highlighting its efficiency and applicability.
>
> ---
>
> **Given the same question, complexity can vary across videos**
>
> This is a great point and is closely related to our discussion of the role of the visual information above. As you mentioned, this limitation arises from the fact that current visual programming methods do not incorporate video information when generating programs. Addressing this, we see two promising directions: (1) adjusting text-based complexity estimation using video features (as we explored in Section 9.1), or (2) incorporating video content directly into the code generation process to improve program accuracy, with the latter approach being more principled. Our results in Section 10.3 in the supplementary demonstrate that advances in visual programming methods directly translate into improved predictive power of code-based metrics, and we expect that incorporating video information in them would significantly enhance our approach.
>
> ---
>
> **Visual programming methods’ performance on NExT-QA and MVBench**
>
> We report ViperGPT’s performance on NeXT-QA in Table 2 in the main paper. As expected, it underperforms compared to the most recent VideoQA models, but still outperforms earlier methods like InternVideo. Note that Section 10.2 shows that even *imperfect* programs produce useful complexity signals. Our proposed CodePlexity metric demonstrates strong robustness to the correctness of the programs used in the analysis, reinforcing its reliability.
>
> Following your suggestion, we have now evaluated ViperGPT and RVP on MVBench, where they achieve accuracies of 38.4% and 35.0%, respectively. For reference, these are comparable to recent approaches such as GPT-4V (run with 16 frames as input at 512×512 resolution), which obtains 43.5%, and VideoChat [1] (also with 16 frames), which achieves 35.5%. This further highlights the relative effectiveness of code-generation-based methods in VideoQA. We will include these results in the final version of the manuscript.
>
> ---
>
> **References:**
>
> [1] Li, KunChang, et al. "VideoChat: Chat-centric video understanding." arXiv preprint arXiv:2305.06355 (2023).
>
> ---

---

> > ### Comment · Reviewer_rEFD · 2025-04-03
> >
> > Thanks for your reply and clarification! I will keep my positive score

---

### Decision · Program_Chairs · 2025-05-01

**Decision:**

Accept (poster)

**Comment:**

This paper proposes a novel data-driven method leveraging visual program generation to systematically evaluate query complexity in VideoQA tasks, introducing CodePlexity as a proxy metric correlating strongly with model performance. Reviewers generally appreciate the methodological novelty, empirical strength, and clear presentation, highlighting that the approach provides valuable insights into model-specific weaknesses and contributes effectively to the creation of challenging benchmarks such as CodePlex-QA. Key concerns raised include reliance on code generation without explicitly considering visual content, potential biases introduced by model-specific definitions of complexity, and generalizability to evolving models. The authors convincingly address these concerns through comprehensive rebuttals, additional experiments, and clarifications, demonstrating robustness, broader applicability beyond VideoQA, and adaptability to newer models. Overall, reviewers agree on the significance and potential impact of this work, leading to a consensus recommendation for acceptance.